# Transcriptional dynamics of murine motor neuron maturation in vivo and in vitro

Tulsi Patel[1] ✉, Jennifer Hammelman[2], Siaresh Aziz[1], Sumin Jang[1], Michael Closser[1], Theodore L. Michaels[1], Jacob A. Blum[3], David K. Gifford[2] & Hynek Wichterle [1] ✉

Neurons born in the embryo can undergo a protracted period of maturation lasting well into postnatal life. How gene expression changes are regulated during maturation and whether they can be recapitulated in cultured neurons remains poorly understood. Here, we show that mouse motor neurons exhibit pervasive changes in gene expression and accessibility of associated regulatory regions from embryonic till juvenile age. While motifs of selector transcription factors, ISL1 and LHX3, are enriched in nascent regulatory regions, motifs of NFI factors, activity-dependent factors, and hormone receptors become more prominent in maturation-dependent enhancers. Notably, stem cell-derived motor neurons recapitulate ~40% of the maturation expression program in vitro, with neural activity playing only a modest role as a late-stage modulator. Thus, the genetic maturation program consists of a core hardwired subprogram that is correctly executed in vitro and an extrinsically-controlled subprogram that is dependent on the in vivo context of the maturing organism.

Advances in directed differentiation of pluripotent stem cells into a growing number of well-defined neuronal cell types have prompted an ever-increasing use of these systems to study normal neuronal function and to model neuronal diseases. However, the ill-defined maturation state of stem cell-derived neurons remains a major shortcoming[1,2]. A thorough understanding of mechanisms that control neuronal specification has been key to the formulation of efficient stem cell differentiation protocols[3,4]. In contrast, our understanding of signals and transcription programs controlling neuronal maturation remains rudimental. In the absence of these mechanistic insights, our ability to assess and manipulate the maturation state of cultured neurons is severely limited.

Most neurons are generated in a brief time window during embryonic development, but the nervous system continues to develop for weeks in mice and years in humans before functional outputs, such as movement and behavior, become adult-like[5,6]. The functional maturation of individual neurons is reflected in gene expression changes that accompany neuronal development from embryonic to adult stages[7–10]. Transplantation studies of human neurons into the mouse brain indicate that neuronal maturation programs have a strong intrinsically timed and species-specific component controlling morphological changes, marker expression, and physiological maturation[11–13]. Besides the intrinsic maturation program, neural activity has been proposed as a key contributor to the maturation of neural circuits in the postnatal brain[14,15]. Indeed, enhancers activated in mature neurons are enriched for motifs recognized by activity-regulated AP-1 transcription factors[9]. While these observations suggest that intrinsic transcriptional programs and extrinsic factors converge to control gene expression programs in maturing neurons, the extent to which these two mechanisms contribute to the maturation process remains a matter of controversy.

Pluripotent stem cell-derived neurons are being widely used to model and study neurodegenerative diseases such as amyotrophic lateral sclerosis (ALS), Alzheimer's, Huntington's, and Parkinson's

[1]Departments of Pathology & Cell Biology, Neuroscience, and Neurology, Columbia University Irving Medical Center, New York, NY 10032, USA. [2]Computer Science and Artificial Intelligence Laboratory, MIT, Cambridge, MA 02139, USA. [3]Department of Genetics, Stanford University School of Medicine, Stanford, CA, USA. ✉e-mail: tp2308@cumc.columbia.edu; hw350@columbia.edu

disease[16]. Neuronal age is the most important modifier in these diseases, as they manifest late in life, even in individuals carrying risk alleles. The immature state of stem cell-derived neurons should therefore be viewed as the key obstacle for meaningful study of these diseases in vitro and for effective drug discovery[1,2]. Recent reprogramming experiments have shown that directly converting adult fibroblasts into neurons, without passing through a pluripotent state, results in the maintenance of some adult-like features, such as DNA methylation, in reprogrammed neurons[17,18]. However, these studies did not directly compare the transcriptome of reprogrammed neurons to their in vivo counterparts, making it difficult to determine the extent to which programmed cells correspond to normal mature neurons.

Motor neurons in the brainstem and spinal cord control all skeletal muscle movements, including walking, breathing, and fine motor skills, by communicating central motor commands to diverse muscle targets. Motor neurons are among the first nerve cells born during embryonic development, but motor circuits and motor behaviors continue to mature over a prolonged period, extending into the postnatal life[6,19,20]. In mice, motor neurons gain their postmitotic identity by embryonic day 10.5 (E10.5); between E10.5 and E13.5 nascent motor neurons start extending their axons toward their cognate muscle targets; between postnatal day 4 (P4) and P21 local spinal interneuron circuits mature, descending motor circuits are wired, and adult-like movement is acquired[4]. Mice are considered sexually mature adults at P56, and aged by year 2. At a genetic level, spinal motor neuron specification is controlled by selector transcription factors ISL1 and LHX3 expressed in nascent postmitotic cells[21,22]. Following initial fate specification, the selector transcription factor LHX3 is downregulated in most motor neurons and the regulatory elements controlled jointly by ISL1 and LHX3 become decommissioned[23], indicating that the maturation program is controlled by additional, yet-to-be-defined transcription factors.

In this work, we mapped transcriptional and chromatin accessibility changes in mouse spinal motor neurons throughout their lifetime to develop a framework for both identifying regulators of maturation and assessing the maturation state of cultured stem cell-derived motor neurons. We demonstrate that the motor neuron gene expression program is highly dynamic for 4 weeks, from embryonic day E10.5 through postnatal day P21, and then remains stable throughout adulthood and aging. By performing motif-enrichment analysis in age-specific chromatin accessibility data, we identified candidate regulators controlling different stages of maturation. The temporal motif-enrichment and expression patterns of candidate regulators suggest a progressive maturation process in which different groups of transcription factors control subsequent stages of motor neuron maturation. Finally, we identified a core maturation program that is faithfully recapitulated in cultured neurons, and demonstrate that despite the commanding presence of motifs bound by activity-dependent transcription factors in regulatory regions associated with these genes, inhibition of action potentials in culture resulted only in minor changes in the later phase of the core maturation gene expression program. This work establishes a foundation for molecular dissection of activity-dependent and independent, intrinsic and extrinsic mechanisms controlling neuronal maturation, a key step toward more reliable use of stem cell-derived neurons for studying normal neural function and dysfunction.

## Results

### Purification of motor neurons for mapping temporal gene expression profiles

The transcriptional identity of a cell is defined by its gene expression profile, which is in turn regulated by the binding of transcription factors at cis-regulatory elements. To understand when a mature motor neuron identity is established, how stably it is maintained, and how it is regulated, we generated a temporal trajectory of gene expression

(RNA-seq) and chromatin accessibility (Assay for Transposase-Accessible Chromatin (ATAC-seq)) data from mouse spinal motor neurons. These data are acquired from embryonic and postnatal motor neurons at timepoints spanning the entire timescale of specification, functional maturation, and aging (E10.5, E13.5, P4, P13, P21, P56 and 2 years; Fig. 1a, Supplementary Fig. 1, and Supplementary Table 1 contains expression data). For gene expression and chromatin accessibility analysis at E10.5, we isolated motor neurons from Hb9:GFP transgenic animals by FACs[24] (Fig. 1b, Supplementary Fig. 1d, e, and Methods). Purification of whole motor neurons from postnatal ages is not feasible as large neurons with complex dendrites and long axons are easily damaged. To overcome this problem, we isolated motor neuron nuclei from Chat-Cre; Sun1-sfGFP-Myc[25,26] mice between E13.5 and 2 years (Fig. 1c, Supplementary Fig. 1, and Methods). We note that using whole cells at E10.5 vs. nuclei at E13.5 onwards may result in technical differences in gene expression values and influence differential gene expression analyses. To determine if these data are comparable, we examined the expression of genes that are known to be up or downregulated between E10.5 and E13.5 and found that these genes maintain the correct dynamics in our RNA-seq dataset[27–29] (Supplementary Fig. 2a).

Another potential limitation of using the Chat-Cre driver from E13.5 to 2 years is that a recent snRNA-seq study found that Chat-Cre labels not only all cholinergic neuronal populations, but also a significant number of GABAergic and glutamatergic neurons in the adult spinal cord[30]. To determine the proportional contribution of individual neural classes to our datasets, we performed a non-negative least squares deconvolution analysis of our bulk adult data (P56), using adult snRNA-seq datasets of spinal cord cell types[30]. Importantly, we did not find any non-cholinergic cells to be significantly represented in our bulk RNA-seq (<1%), perhaps reflecting a difference in INTACT (this study) vs. FACs methods for purifying nuclei for RNA-seq (Supplementary Figs. 1b–d and 2b–d). Furthermore, the analysis revealed that the vast majority (~76%) of bulk P56 gene expression comes from skeletal motor neurons, with ~22% contribution from cholinergic visceral motor neurons and ~2% from cholinergic interneurons (Supplementary Fig. 2d, e). This is further supported by the analysis of expression levels of cell-type-specific transcription factors and effector genes in our dataset[29–40]. Whereas skeletal motor neuron-specific genes (e.g., *Isl1*, *Ebf1*, *Onecut2*, *Stat2*, *Klf6*, and *Foxp1*) are highly expressed, visceral motor neuron, interneuron, and non-cholinergic interneuron genes[31] are expressed at lower levels or not expressed at all (mean expression value: skeletal 559, visceral 82, interneuron 99) (Fig. 1d and Supplementary Fig. 2f, g). We conclude from this analysis that skeletal motor neurons overwhelmingly contribute to the gene expression profiles, with some contribution from cholinergic visceral motor neurons. For ATAC-seq we utilized FAC sorted SUN1-GFP nuclei (Methods) and confirmed that all candidate regulators identified from this analysis are expressed in skeletal motor neurons (Supplementary Fig. 1e).

### Motor neuron gene expression is dynamic until the third postnatal week

We first asked when an adult-like gene expression pattern is established in motor neurons and how stably it is maintained. To answer this question, we performed a principal component analysis on the temporal RNA-seq data (Fig. 1e). We found that the first principal component (PC1), which explains 53.2% of the overall variance, separates the gene expression data from E10.5 to P21 sequentially by age, with all timepoints after P21 clustering together. Next, to understand what proportion of motor neuron genes are dynamically regulated between timepoints, we performed differential gene expression analyses using the EdgeR algorithm[41,42] (Fig. 1f, g). These analyses show that >40% of expressed genes are differentially expressed between E10.5 and E13.5, and between E13.5 and P4 (*p* value <0.001, fold change ≥2). After P4,

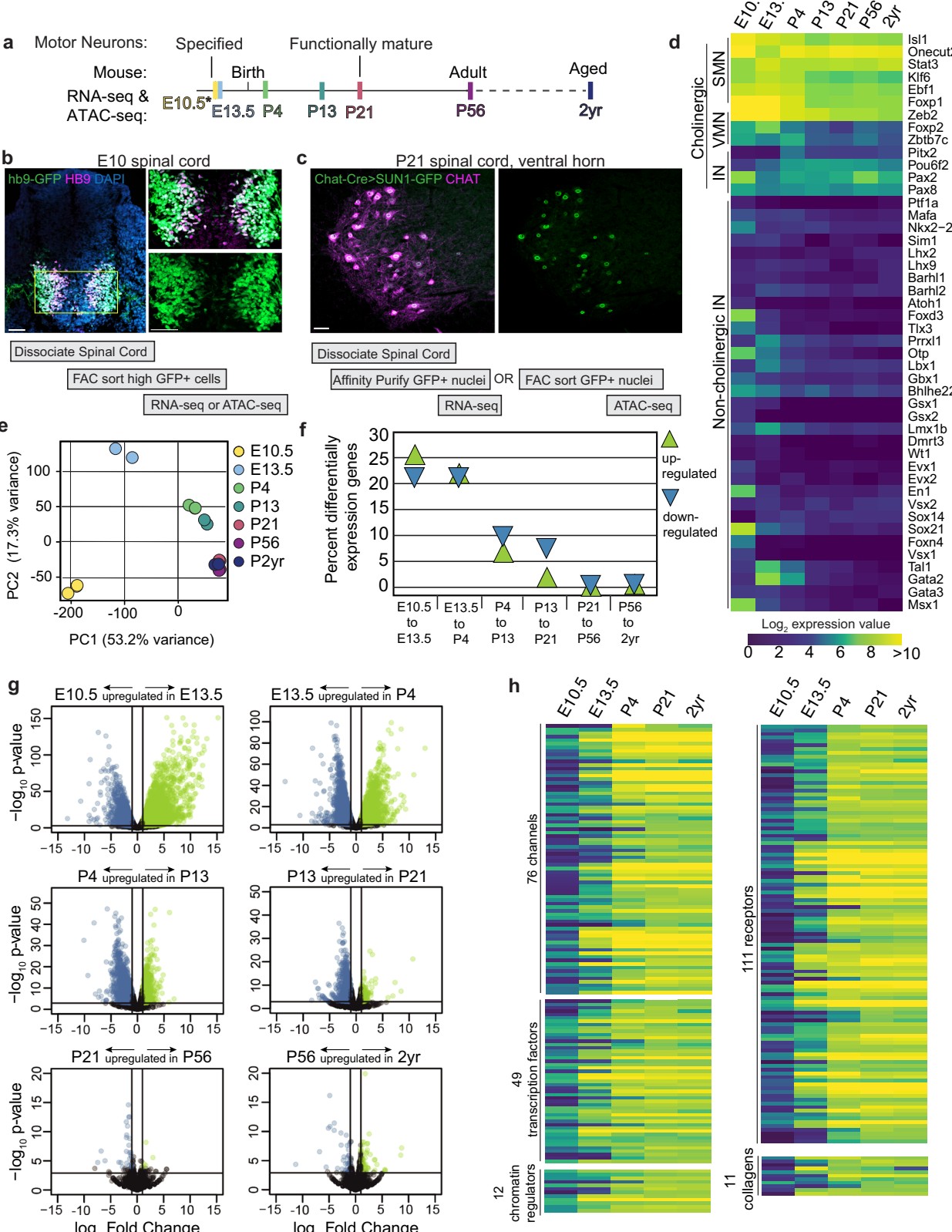

both the percent of genes that are differentially expressed and the extent of the differential expression (in terms of $p$ value and fold change) decreases as motor neurons get older, with <1% of genes changing expression between P21 and P56 and between P56 and 2 years (Fig. 1f, g). The RNA-seq data reveal a striking concordance between the timing of transcriptional maturation and functional

maturation of motor circuits and behavior, which have previously been shown to happen during the third postnatal week, between P14 and P21[2,6,19]. This correlation suggests that transcriptional changes may facilitate the functional output of adult motor neurons. Indeed, when we perform a pathway enrichment analysis[43] on the top 500 genes that are upregulated between E10.5 and P21 or E13.5 and P21, we find that

**Fig. 1 | Motor neuron gene expression is dynamic during embryonic and early postnatal ages. a** Timeline of motor neuron development and functional maturation in mice. **b** A spinal cord section from an Hb9-GFP transgenic E10 embryo and outline of the procedure used to isolate motor neurons. Immunostaining shows co-labeling of HB9 protein (magenta) with high GFP expression from the transgene (green). The boxed region from the left image is shown with split colors on the right. **c** The right ventral horn region of a P21 Chat-Cre; Sun1-GFP spinal cord section and outline of the procedure used to isolate motor neuron nuclei at all ages from E13.5 to 2 years. Immunostaining shows co-labeling of CHAT protein (magenta) with SUN1-GFP (green). **d** Expression of transcription factors that mark cholinergic skeletal motor neuron (SMN), visceral motor neurons (VMN),

interneurons (IN), and non-cholinergic cell types in the spinal cord at all profiled ages. A handful of VMN and IN markers are also expressed in SMNs, as seen in Supplementary Fig. 2f. **e** Principal component analysis on RNA-seq data at all timepoints. Circles of the same color represent biological replicates for each timepoint. **f** Percent of expressed genes that are upregulated or downregulated between two consecutive timepoints. This plot is a summary of the differential gene expression plots in (**g**). **g** Plots showing differential gene expression between consecutive timepoints. Each dot represents one gene, colored dots are genes that are upregulated (green) or downregulated (blue) at least 2-fold with a *p* value <0.001. **h** Expression trajectories of various categories of functional genes. The scale for the heatmap is the same as in **d**. All scale bars are 50 μm.

most enriched pathways include categories relevant for motor neuron physiology such as protein interactions at synapses, and muscle contraction (*p* value <0.001, FDR <0.05). On the other hand, genes downregulated with age are enriched in shared cellular pathways characteristic of mitotically active or growing cells, such as RNA and protein metabolism, translation, and mitotic pathways, in addition to pathways associated with early stages of neurodevelopment such as axon guidance (*p* value <0.0005, FDR <0.05) (Supplementary Fig. 3a).

We next wanted to understand the extent to which select groups of neuronal effector genes were dynamically expressed during maturation. Analysis of differentially expressed genes showed that 55% of channels, 32% of receptors, 50% of collagens, 18% of kinases, 15% of transcriptional regulators, and 10% of chromatin regulatory genes expressed in our dataset show low or no expression at E10.5, but are upregulated at least 2-fold by P21 (*p* value <0.001, subsets shown in Fig. 1h, full lists in Supplementary Table 2). We confirmed the postnatal upregulation of functionally relevant genes such as sodium and potassium channels (*Scn4b, Kcng4, Kcnj14*), a glutamate receptor (*Grin3b*), enzymes (*Aox1, Gls2, Rab37*), structural proteins (*Tmod1, Dmp1, Col5a3*), a secreted phosphoprotein (*Spp1*), and thyroid hormone-related factors (*Thrsp, Thrb*) using Allen Brain Atlas[44] in situ hybridization data (Supplementary Fig. 4). Several of these genes have been implicated in motor neuron function: *Kcng4*[45] and *Kcnj14*[46] modulate firing properties of motor neurons, *Scn4b* mutant animals show motor defects[47], *Grin3b* is a motor neuron-specific NMDA receptor subunit that forms excitatory glycinergic receptors[48], expression levels of *Aox1, Col5a3*, and *Gls2* are altered in mouse models of a motor neuron disease ALS[49–51], and *Spp1* shows reduced expression in motor neurons and increased levels in cerebrospinal fluid of ALS patients[52–54]. We also noted that proteasome and TriC/CCT chaperonin complex genes are significantly downregulated during the transition from embryonic to postnatal state, which may contribute to the inability to clear misfolded proteins in ALS[55] (Supplementary Fig. 3b). Thus, genes involved in broad aspects of neuronal identity, function, and pathology are dynamic during postnatal motor neuron maturation.

## Genes upregulated during maturation are largely cell-type specific and contribute to functional diversification

Recent gene expression analyses performed in cortical neurons[9], somatosensory neurons[8], and hypothalamic neurons[7] demonstrate that many, if not all, postmitotic neurons undergo a protracted period of transcriptional maturation. We wondered whether gene expression changes in postnatal neurons reflect a shared or cell-type-specific maturation program. To answer this question, we took advantage of comparable gene expression data obtained from nuclei of cortical VIP and SST inhibitory interneurons and RORB-expressing excitatory neurons from postnatal ages P7–P56[9]. To define broadly shared aspects of the neuronal maturation program, we selected genes that are differentially regulated in motor neurons between P4 and P56 (1004 up, 1742 down genes; *p* value <0.001, fold change ≥2), and asked what percent of these genes are regulated in the same direction in VIP, SST, and RORB neurons (with *p* value <0.001 in all three cell types). We

found that only 27% (276) of motor neuron upregulated genes are also upregulated in the three cortical cells, whereas 54% (948) of downregulated genes are shared (Supplementary Fig. 5a). When we perform pathway enrichment analysis[43] on the shared downregulated genes, the most enriched pathways are those related to protein metabolism, such as peptide chain elongation (65 genes), rRNA processing (66 genes), and pathways related to neural development such as axon guidance (135 genes; *p* value <1e−16, FDR <7e−15 for all). The shared upregulated genes, on the other hand, were only enriched for one pathway, potassium channels (11 genes, *p* value 1.84e−5, FDR 0.012). We conclude from this analysis that diverse neuron types share about half of the downregulated program during maturation, which includes metabolic and developmental genes. A majority of genes upregulated during maturation, however, are cell-type specific and likely contribute to the diverse adult functions of each neuron type.

The cell-type specificity of postnatal maturation led us to wonder if functional diversification within motor neurons occurs during this time. Embryonic spinal motor neurons are categorized into column and pool subtypes based on the muscle groups they innervate. Mature motor neurons can be further subdivided functionally into alpha and gamma types based on their electrophysiological properties[56,57]. Recent single nucleus analyses of adult motor neurons identified several markers that distinguish motor pools and alpha vs. gamma motor neurons[30,31]. Interestingly, we found that out of 28 pool identity markers[31], ~83% are upregulated between E10.5 and E13.5 and only 7% are upregulated postnatally between P4 and P21 (Supplementary Fig. 5b, c). On the other hand, markers that distinguish alpha motor neurons from gamma motor neurons[30,31] continue to be upregulated in postnatal life. Of 25 alpha markers, 28% are induced or upregulated between E10.5 and E13.5 and 48% are upregulated between P4 and P21 (Supplementary Fig. 5d, e). These data show that gene expression programs that define electrophysiologically divergent alpha motor neurons continue maturing during the postnatal period; in contrast, differences among individual motor pools become crystalized early during embryonic development when pool-specific gene expression programs are required for accurate muscle innervation[27,58].

## Changes in chromatin accessibility accompany dynamic gene expression

Having determined the temporal dynamics of gene expression, we next sought to identify the regulatory regions that orchestrate maturation. Accessible chromatin regions are known to harbor binding sites for regulatory transcription factors and can be effectively mapped by ATAC-seq[59–62]. To identify putative regulators of neuronal maturation we performed ATAC-seq at E10.5, E13.5, P4, P13, P21, P56, and 2 years. Principal component analysis demonstrated that similar to gene expression changes, regulatory regions are dynamic between E10.5 and P21, with all the remaining postnatal ages clustering together (Fig. 2a). To understand the dynamics in more detail, we asked what percent of the top 100k accessible regions at each age either gain or lose accessibility compared to adjacent timepoints. We found that about half of all peaks are dynamic between E10.5 and E13.5 and >20% of peaks are dynamic between E13.5 and P4 and between P4 and P13

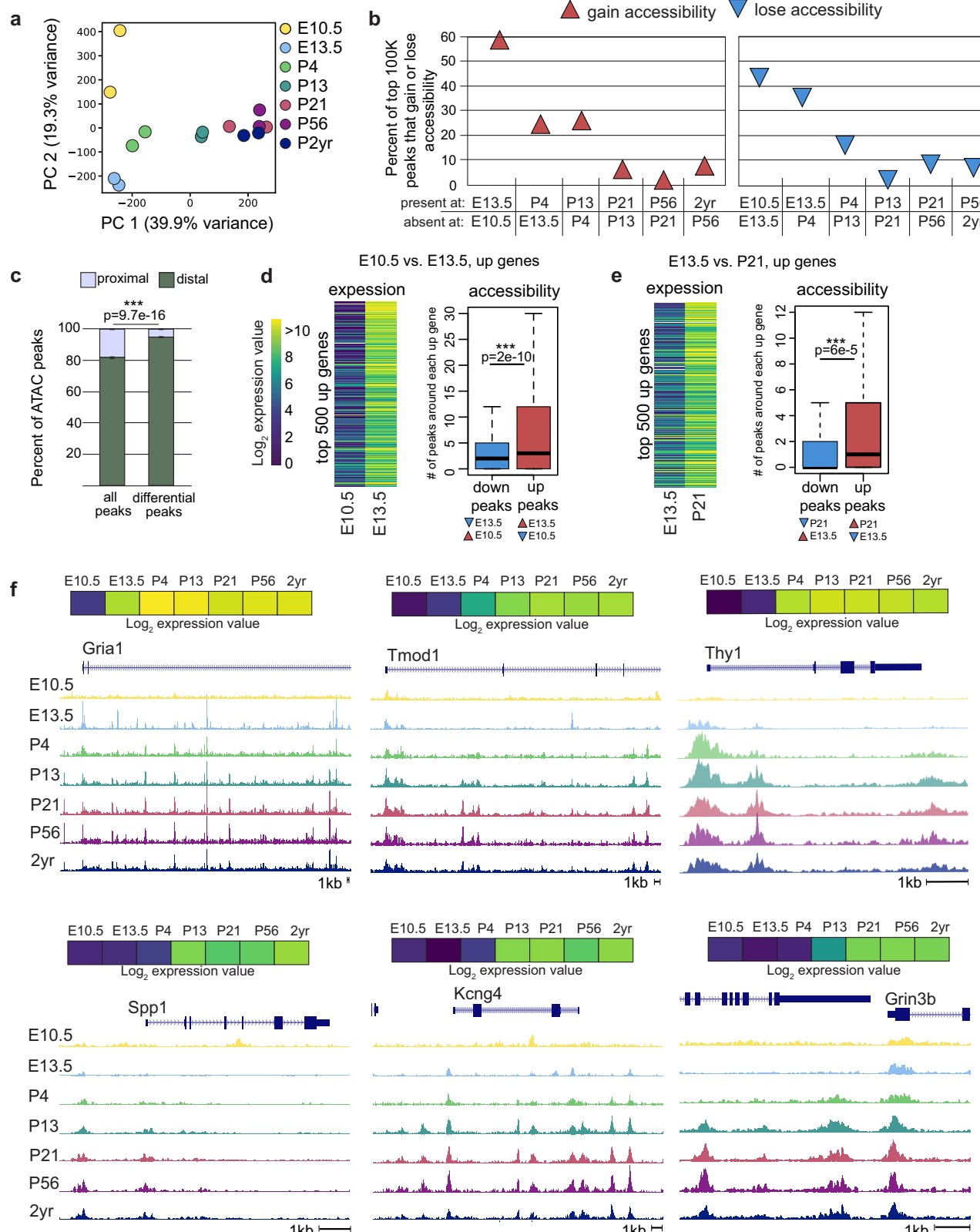

(Fig. 2b). The vast majority of differentially accessible chromatin regions (~95%) are located at least 2 kb away from transcription start sites (Fig. 2c), consistent with previous findings that dynamic changes in gene expression are regulated by distal regulatory regions rather than promoter proximal ones[9,63,64].

To examine whether changes in chromatin accessibility correlate with the trajectory of gene expression changes during motor neuron maturation, we examined the dynamic peaks around 500 upregulated and 500 downregulated genes at two temporal transitions: E10.5–E13.5, and E13.5–P21. At both temporal transitions, we found that the upregulated genes are surrounded by a significantly higher number of newly accessible distal peaks than peaks that lose accessibility (Fig. 2d–f). Moreover, of the top 500 genes induced during both temporal transitions, the majority are associated with at least one new

**Fig. 2 | Dynamic changes in accessible regions accompany gene expression changes. a** Principal component analysis on ATAC-seq data at all timepoints. Circles of the same color represent biological replicates. **b** The percent of top 100k genomic regions that lose accessibility or become newly accessible between consecutive timepoints. **c** The percent of total peaks or differential peaks that are proximal (within 2 kb) or distal (>2 kb away) from transcription start sites. Error bars represent SEM and p value is from two-tailed t-tests, n = 7 (all peaks), 12 (differential peaks). **d**, **e** The distribution of differentially accessible regions near genes that are upregulated during maturation between E10.5 and E13.5, and E13.5 and P21. The gene expression heatmaps show the expression values of top 500 upregulated genes. The box plots show the number of peaks around each of the 500 upregulated genes that become newly accessible (red) or lose accessibility (blue) between E10.5 and E13.5, and E13.5 and P21. Center line of box plots is the median, the interquartile range is 25th–75th percentile, and outliers are eliminated; p values are from two-tailed t-tests; n = 500 genes. **f** ATAC-seq reads around upregulated genes. Source data are provided as a Source Data file.

accessible chromatin peak (74% for E10.5–E13.5, and 63% for E13.5–P21). In contrast, downregulated genes are surrounded by equal numbers of gained and lost peaks (Supplementary Fig. 6a, b). Altogether, these data show that upregulated genes are preferentially surrounded by newly accessible peaks that may control their maturation-dependent expression.

## Motif analysis identifies putative regulators of maturation

To identify putative regulators of maturation, we performed motif-enrichment analysis on accessible regions using two distinct but complementary approaches. We used a newly developed convolutional neural network-based method called DeepAccess[65,66] which learns the relationship between primary DNA sequence and chromatin accessibility at different ages and then predicts transcription factor motifs that drive differential chromatin accessibility at one age compared to another. In addition, we used HOMER, a de novo motif discovery method that identifies enriched motifs in provided sets of age-specific accessible peaks[67] (Methods). Together these two methods allow us to identify motifs that are correlated with global accessibility in maturing neurons, and motifs that are enriched in peaks that gain or lose accessibility over time.

HOMER motif-enrichment analysis of top 10k peaks (Methods) at E10.5 and E13.5 identified several transcription factor families that are already known to regulate gene expression in motor neurons: Sox, bHLH (bind E-box), Retinoid receptors (bind RXR), Hox, LIM, and Onecut[22,23,68,69] (Fig. 3b and Supplementary Fig. 6c). However, analysis of later ages shows that many peaks containing these motifs are decommissioned over time and new transcription factor families become more prominent (Fig. 3b–d and Supplementary Fig. 7).

Both DeepAccess and HOMER identified an overlapping set of four candidate regulators of motor neuron maturation: NFI transcription factors, steroid hormone receptors (bind to GRE motifs), AP-1, and Mef2 transcription factors (Fig. 3a–d and Supplementary Fig. 7). NFI motifs are enriched at all ages after E10.5 in both DeepAccess and HOMER analysis. Not only do these motifs become highly enriched during the transition from E10.5 to E13.5, but also they continue to be enriched in peaks gaining accessibility at P4 and at P21. This suggests that NFI factors bind different sites at different stages of maturation, perhaps by cooperating with temporally regulated cofactors. Indeed, when we search for secondary motifs, we find that NFI containing E13.5 > E10.5 peaks are enriched for HOX and LIM motifs, whereas NFI containing peaks at P21 > P4 are enriched for AP-1 motifs (Fig. 3e). At postnatal ages after P4, we observe an enrichment of steroid hormone receptor, Mef2, and AP-1 motifs in addition to NFI. Steroid hormone receptors are activated by steroid hormone ligands[70], and both Mef2 and AP-1 factors are activated in response to calcium influx in neurons firing action potentials[71,72].

We next identified transcription factors from these families that are expressed in motor neurons using the RNA-seq dataset and immunostaining (Fig. 3f, g and Supplementary Fig. 6d). Consistent with our motif-enrichment analysis, NFI family genes *Nfia* and *Nfib* are highly induced in motor neurons between E10.5 and E13.5, and remain high throughout life, as has been previously reported (Fig. 3f, g)[29,73,74]. Notably, expression of NFIA and NFIB proteins is low or absent in a small subset of motor neurons, likely corresponding to gamma motor neurons based on the smaller size of their cell bodies and the fact that

*Nfia/b* are expressed lower in gamma motor neurons in snRNA-seq data (Supplementary Fig. 6e). Among activity-dependent transcription factors, we detected continuous expression of *Mef2a*, *Mef2d*, and *Jun*, but very low expression of *Fos* (Fig. 3f). *Fos* transcripts may not be effectively detected in bulk nuclear RNA, as this immediate early gene is only transiently and rapidly induced in response to activity[75]. Indeed, immunostaining of spinal cords for FOS protein revealed expression in subsets of postnatal spinal motor neurons. The proportion of neurons that express FOS increased from ~20% at P4 to ~80% at P56 in ventral horn motor neurons, in agreement with the higher enrichment of activity-dependent motifs after P4 (Fig. 3g). And finally, we detect the expression of steroid hormone receptors, *Nr3c1* (glucocorticoid receptor), *Nr3c2* (mineralocorticoid receptor), *Pgr* (progesterone receptor), and *Ar* (androgen receptor) (Fig. 3f, extended Fig. 6d). Immunostaining revealed that both NR3C1 and NR3C2 are expressed in nearly all adult skeletal motor neurons (Fig. 3g). While *Nr3c1* and *Nr3c2* mRNA is increased already by E13.5 and P4, respectively; proteins only become visible in postnatal P4 and P56 motor neurons, respectively, consistent with the timing of their motif enrichment in the ATAC-seq data. (Fig. 3g). Together, these data suggest that neuronal maturation is regulated by a progressive series of temporally regulated cell intrinsic transcription factors, neuronal activity, and extrinsic signaling molecules.

## Long-term cultures of stem cell-derived motor neurons partially recapitulate in vivo maturation

Stem cells can be effectively differentiated into defined neuron types, including spinal motor neurons[24]. However, differentiated motor neurons adopt an immature embryonic-like state[76] and it remains unclear to what extent they can undergo in vivo-like maturation changes in the absence of relevant synaptic partners, muscle targets, and other signaling factors like circulating hormones. Our in vivo profiling revealed that motor neurons in the spinal cord take about ~4 weeks (from E10.5 to P21) to become transcriptionally mature. We wondered whether stem cell-derived motor neurons could also mature over the same timescale when co-cultured with astrocyte-enriched primary glial cells, which have been shown to promote neuronal survival, synaptogenesis, and electrophysiological maturation[77,78]. To test this, we differentiated Chat-Cre; Sun1-sfGFP-Myc embryonic stem cells[79] into motor neurons, using an established differentiation protocol that produces motor neurons of cervical identity[80] and co-cultured them on primary astrocytes for 28 days (Fig. 4a, b and Methods). We noted that Sun1-sfGFP-Myc positive motor neurons retain expression of ISL1/2 in co-culture, get larger, and elaborate complex neurites over time (Fig. 4c).

To determine if motor neurons undergo gene expression changes in culture, we performed RNA-seq analysis of purified motor neuron nuclei at DIV0, DIV7, and DIV28 (Supplementary Table 1 and Supplementary Fig 8a). Both principal component analysis and differential gene expression analysis showed that gene expression of cultured motor neurons is dynamic over time (Fig. 4d and Supplementary Fig. 8b). In order to understand if the genes that change between DIV0 and DIV28 are relevant to in vivo maturation, we performed an integrated principal component analysis on these genes across all in vivo and in vitro timepoints. This analysis shows DIV0 aligning with E13.5, DIV7 with P4, and DIV28 with mature P21–2 years on PC1, suggesting

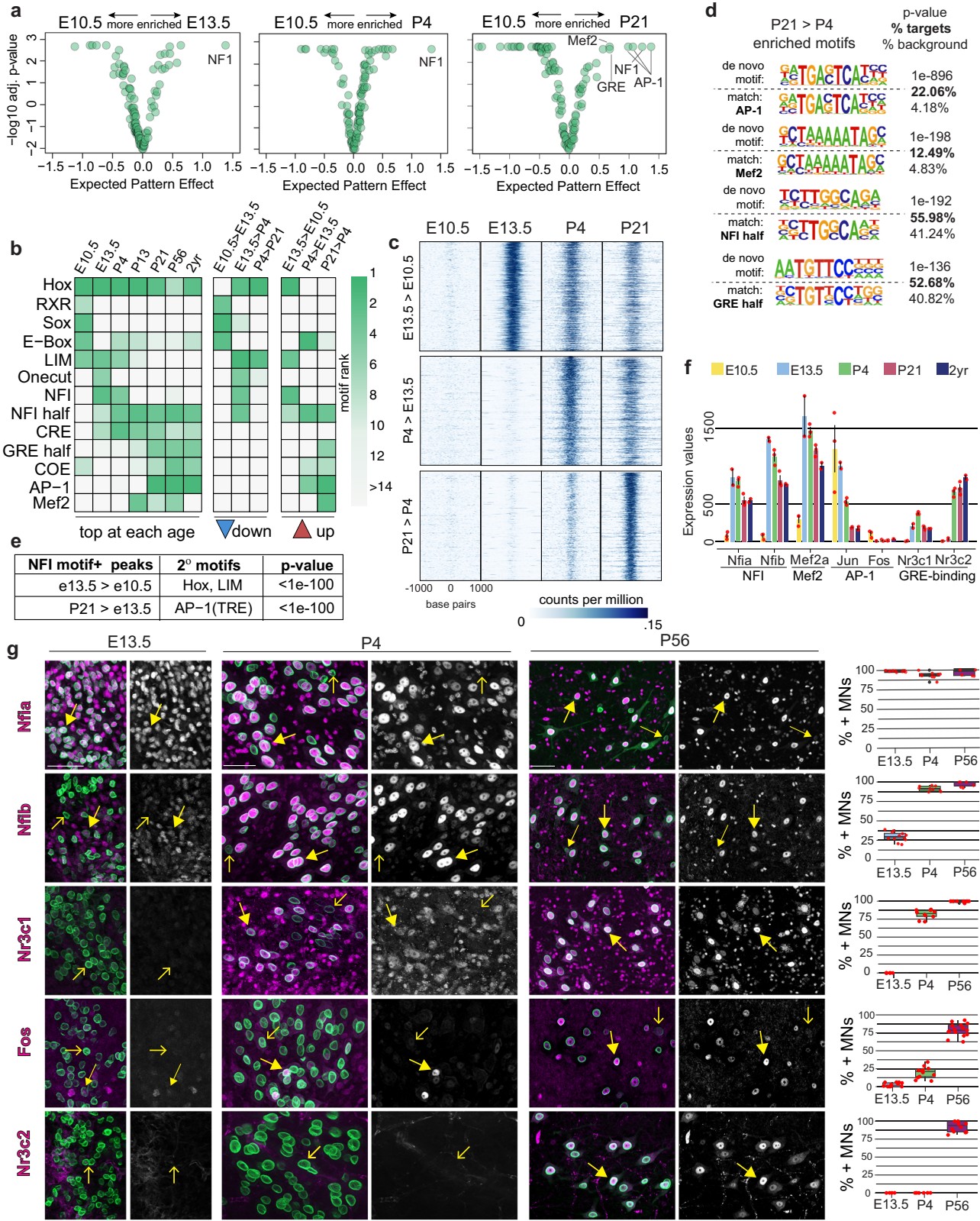

that genes loading the first component faithfully recapitulate in vivo-like transitions in cultured motor neurons (Fig. 4e).

To further investigate the correlation between in vitro and in vivo expression changes, we plotted the in vitro vs. in vivo log fold change for all expressed genes (Fig. 4g). This analysis showed a positive correlation between in vitro and in vivo transcriptional maturation (correlation: $R = 0.45$, $p$ value <2.2e−16), with ~40% of genes that are

upregulated and ~30% of genes that are downregulated >2-fold in vivo recapitulating the same dynamics in vitro. Interestingly, when we perform a global principal component analysis based on expression of all genes, we see that in vitro motor neurons are most similar to P4 cells, showing only minor shifts toward more mature in vivo states (Supplementary Fig. 8c). This is likely because ~55% of genes that are dynamic during in vivo maturation remain unchanged in cultured

**Fig. 3 | Identification of putative regulators of motor neuron maturation.**
**a** Identification of motifs enriched during maturation using DeepAccess. Each dot represents one of 108 consensus transcription factor binding motifs. The expected pattern effect (*x*-axis) is a score of how differentially predictive each motif is of accessibility between two timepoints. For example, NFI motifs are highly predictive of accessibility at E13.5, P4, and P21 as compared to E10.5. **b** Motif enrichment using HOMER. Each column shows the most highly enriched motifs in the given condition. The left panel shows the most highly enriched motifs in the top 10k regions at each age. The two right panels show the most highly enriched motifs in the top 10k differentially accessible peaks. **c** ATAC-seq reads at 10k most significant genomic regions that are newly accessible between E10.5 and E13.5, E13.5 and P4, and P4 and P21. Each panel spans ±1 kb from the center of the peak. **d** HOMER outputs of motifs enriched in top 10k newly accessible peaks between P4 and P21. The motif logos show the de novo motif (top) and the best-matched known transcription factor motif (bottom) along with *p* value and prevalence. **e** Secondary motifs found in NFI-motif-containing peaks using HOMER, *p* value generated by HOMER. **f** Expression values of select transcription factors that bind to enriched motifs. Expression of all transcription factor family members is shown in Supplementary Fig. 6d. Error bars show SEM; *n* = 3 biological RNA-seq replicates for all except E13.5 and P13 (*n* = 2), individual replicates shown as red dots. **g** Immunostaining and scoring of candidate regulators in E13.5, P4, and P56 spinal cords. Full arrows point to cells that are scored as positive (large arrows-bright, small arrows-dim), empty arrows point to negative cells. Center line of box plots is the median, the interquartile range is 25–75th percentile; *n* = 3 independent animals, multiple sections scored and all individual data points are shown as red dots. Scale bars are 50 μm. Source data are provided as a Source Data file.

motor neurons, and these genes show P4-like levels of expression at all in vitro timepoints (Fig. 4f, g, Supplementary Fig. 8d, e, and Supplementary Table 3). Thus, our data suggest that the in vivo motor neuron maturation program can be subdivided into a core set of genes that faithfully recapitulates the temporal maturation dynamics in vitro and the second set of genes that is dependent on the in vivo context for their proper regulation.

The core set of maturation genes that exhibit proper regulation in vitro consists of 2796 genes (Fig. 4g). Not only do these genes change in the same direction both in vitro and in vivo, but also they show no significant difference in expression levels between DIV28 and P21 motor neurons (Fig. 4h, i). We validated the in vitro expression data for two genes: *Thy1* using a reporter stem cell line (Methods), and *Spp1* using immunostaining. These genes show different temporal dynamics in their upregulation in vivo. While *Thy1* is upregulated in the embryonic to early postnatal transition, *Spp1* is upregulated later during postnatal life. In vitro analysis of reporter and protein expression shows that these temporal dynamics are recapitulated in the culture system—*Thy1* expression is upregulated between DIV7 and DIV14, whereas *Spp1* expression is upregulated later between DIV14 and DIV28 (Fig. 4j, k).

We next asked if genes that are shared in vivo and in vitro are functionally relevant for motor neuron maturation. Pathway enrichment analyses[43] showed that co-upregulated genes are enriched for neuronal pathways such as potassium channels and protein interactions at synapses (Supplementary Fig. 8g). We observe that 43% of kinases, 48% of receptors, and 57% of channels that are upregulated in vivo are also upregulated in vitro (Supplementary Fig. 8f). Co-downregulated genes are enriched for early developmental pathways like cell cycle, activation of HOX, Notch and Wnt signaling (Supplementary Fig. 8h). We also used SynGo[81], a curated database of synaptic genes, to perform term enrichment on genes upregulated between E10.5 and P21 and between DIV0 and DIV28. We found that the same categories of synaptic genes are upregulated in both gene sets, suggesting that the upregulation of synaptic genes in culture is similar to in vivo (Supplementary Fig. 8i). Lastly, we wondered if the maturation subprogram that is recapitulated in vitro is enriched for genes that are generically upregulated in all maturing neuron types. Interestingly, based on the comparison between motor neurons and cortical neurons (Supplementary Fig. 5a), the core maturation - upregulated genes are just as likely to be motor neuron-specific as all genes upregulated in vivo (70 vs. 73%). The 2796 shared genes therefore constitute a largely motor neuron-specific maturation program, consisting of functionally relevant effector genes contributing to the physiological maturation of motor neurons both in vivo and in vitro.

### Chromatin accessibility identifies putative shared and in vivo-specific regulators of maturation

Having established transcriptional similarities between in vivo and in vitro maturation, we next asked if in vitro neurons employ in vivo-like regulatory mechanisms to control temporal gene expression changes by performing ATAC-seq on stem cell-derived motor neurons at DIV0, DIV7, and DIV28. As is the case in vivo, we identified over 100k accessible regions at all in vitro timepoints, found that accessible peaks are highly dynamic in culture, that differential peaks are disproportionately distal, and that the top 10k peaks at each timepoint are more conserved than random intergenic regions, suggesting their functional relevance (Supplementary Fig. 9a–d). Importantly, when we performed a principal component analysis using all distal ATAC peaks in both in vivo and in vitro timepoints, we found that PC1 separates the data by time, with DIV0 aligning closer to embryonic ages, and DIV28 to more mature P13/P21 motor neurons (Fig. 5a, b). This suggests that changes in global chromatin accessibility in vitro are better correlated with in vivo maturation than global gene expression patterns, which showed the in vitro timepoints clustered around P4 (Supplementary Fig. 8b).

The identification of in vitro accessible chromatin regions provided an opportunity to identify putative regulators of the shared and in vivo-specific maturation subprograms. We performed motif-enrichment analysis on distal genomic sites that are associated with maturation-dependent genes and that gain accessibility over time both in vivo and in vitro (between E10.5 and P21 in vivo and between DIV0 and DIV28 in vitro). We found that these sites show a high prevalence for AP-1 and NFI motifs, with activity-dependent AP-1 motifs enriched at a much higher significance than any other motif (Fig. 5c, top panel and Fig. 5d). Motif-enrichment analysis on the top 10k significant peaks that are upregulated globally between DIV0 and DIV28 in culture also shows high enrichment of AP-1 motifs, and moderate enrichment of NFI motifs (Supplementary Fig. 9e). Accordingly, we observed robust expression of immediate early gene FOS, and NFI transcription factors NFIA and NFIB in DIV28 motor neurons (Fig. 5f, g).

To identify potential in vivo-specific regulators, we performed motif analysis on in vivo-specific differential peaks around in vivo-specific upregulated genes. This analysis identified Hox motifs and steroid hormone receptor motifs in addition to NFI motifs (Fig. 5c, bottom panel and Fig. 5e). Importantly, we have not detected enrichment for AP-1 motifs in in vivo-specific accessible regions, indicating that most activity-dependent chromatin changes are recapitulated in cultured stem cell-derived motor neurons. Together these data suggest that, despite the differences in neural circuits formed in vivo and in vitro, activity and NFI factors are important contributors to motor neuron maturation in both systems. In vivo motor neuron maturation appears to be additionally controlled by Hox factors and hormonal signaling in postnatal animals[82].

### Neuronal activity plays a limited role as a modulator of the later stage maturation program in vitro

Since activity-dependent AP-1 motifs are highly enriched in maturation-dependent accessible chromatin regions in vitro (Fig. 5d and Supplementary Fig. 9e), we used the culture system to investigate the extent to which neuronal activity contributes to the motor neuron maturation program. Action potentials can be effectively blocked in

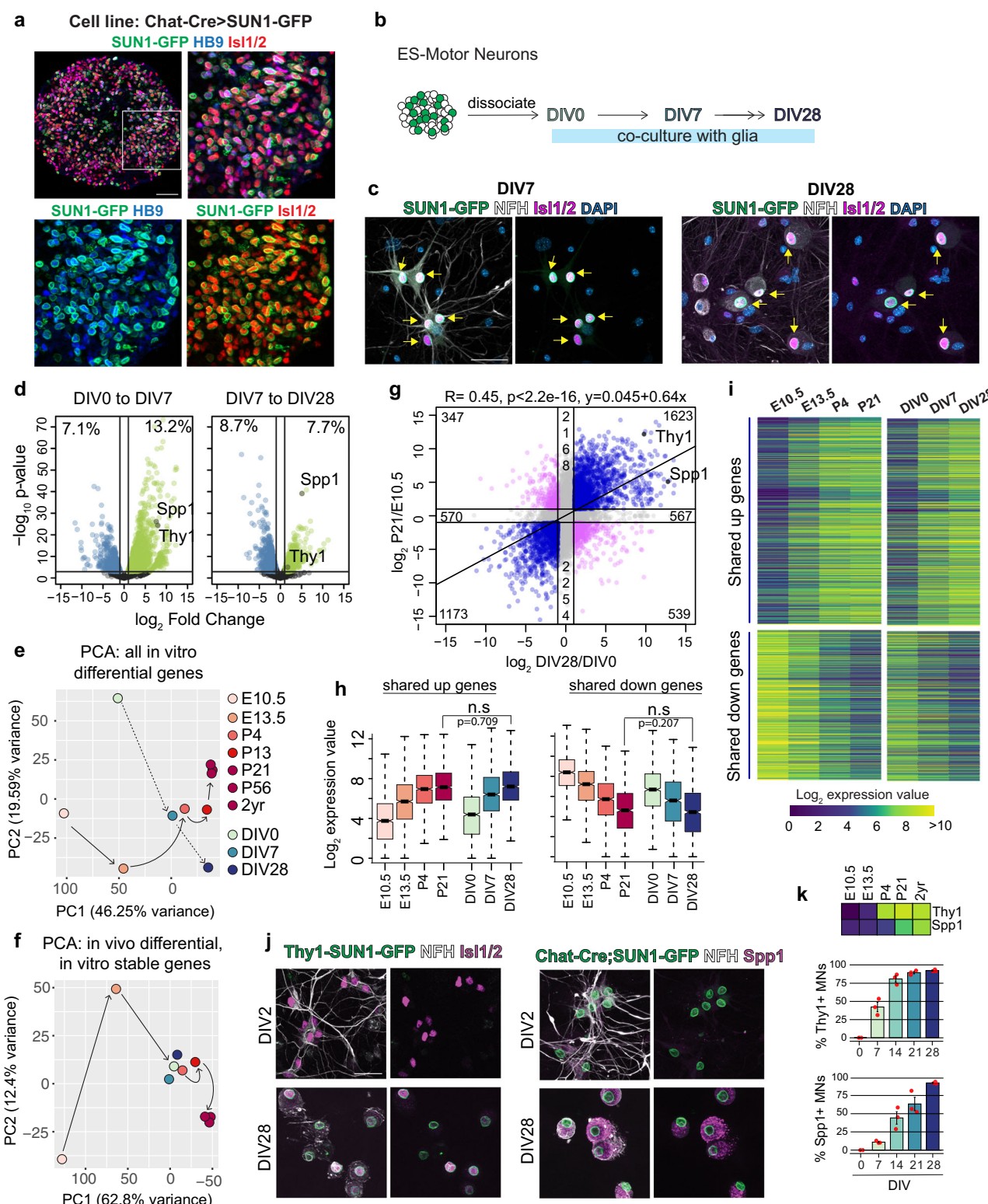

culture by growing cells in the presence of sodium channel inhibitor tetrodotoxin (TTX), leading to a suppression of the immediate early gene FOS (Fig. 5g). We performed ATAC-seq and RNA-seq on motor neurons grown in the presence of TTX from DIV0 until DIV7 or DIV28, to test the early vs. late role of neuronal activity (Fig. 6a).

Analysis of DIV7 data shows that loss of activity has minimal effect on both chromatin accessibility and gene expression during the first week of maturation. We identified no differentially accessible peaks between DIV7 + TTX vs. DIV7 motor neurons (FDR <0.5), and only 13

differentially expressed genes at this timepoint (p value <0.001, fold change >2) (Supplementary Fig. 10a, b). In contrast, activity regulates maturation-dependent transcription to a greater extent during the transition between DIV7 and DIV28. We identified 2409 ATAC peaks differentially accessible[83] (FDR <0.05) between DIV28 and DIV28 + TTX (Fig. 6b). Two-thirds of these peaks show reduced accessibility in TTX (1674 out of 2409), and the top two motifs enriched in these down-regulated peaks are of activity-dependent AP-1 and Mef2 factors (Fig. 6c). Furthermore, majority of these downregulated peaks are not

**Fig. 4 | Stem cell-derived motor neurons undergo maturation-relevant gene expression changes. a** Image of embryoid body differentiated from Chat-Cre; Sun1-GFP stem cell line showing co-expression of SUN1-GFP (green) with motor neuron markers ISL1/2 (red) and HB9 (blue). Boxed region on the top left is expanded in the remaining three panels. **b** Schematic of in vitro maturation experiments: motor neurons are co-cultured with astrocyte-enriched glial cells. **c** SUN1-GFP expressing motor neurons stained with motor neuron marker ISL1/2 (magenta), and Neurofilament heavy chain (NFH in white) at DIV7 and DIV28 of co-culture. Arrows point to SUN1-GFP+ ISL1/2+ motor neurons. **d** Plots showing differential gene expression between consecutive in vitro timepoints. Each dot represents one gene, colored dots are genes that are upregulated (green) or downregulated (blue) at least 2-fold with a *p* value <0.001, as determined by EdgeR. The percent of expressed genes that are dynamic is also reported. **e** Principal component analysis on expression of in vitro differential genes (>2-fold change between DIV0 and DIV28) at all in vivo and in vitro timepoints. Each dot is the

average expression of all biological replicates. **f** Principal component analysis on expression data from in vivo dynamic, in vitro stable genes (2168 + 2254 genes from **g**), labels same as **e**. **g** Correlation between changes in gene expression in vitro and in vivo. Each dot represents a gene, and its placement on the plot shows how its expression changes between DIV0 and DIV28 in vitro (*x*-axis) and between E10.5 and P21 in vivo (*y*-axis). Blue genes have shared regulation while pink genes have opposite regulation in vitro and in vivo. The total number of genes in each category is also reported. **h, i** Expression of shared up- and downregulated genes. Center line of box plots is the median, the interquartile range is 25–75th percentile, and outliers are eliminated; n.s. *p* value >0.05 as determined by two-tailed *t*-tests; *n* = 1623 up and 1173 down genes. **j, k** Images and scoring of Thy1-P2A-SUN1-GFP reporter and SPP1 in cultured motor neurons, as detected by immunostaining, and heatmap showing expression of *Thy1* and *Spp1* in vivo. Error bars show SEM; *n* = 3 independent biological replicates shown as red dots. All scale bars are 50 μm. Source data are provided as a Source Data file.

constitutive accessible regions, but gain accessibility during in vitro maturation (Fig. 6d). These analyses cumulatively show that activity-dependent transcription factors are required to increase accessibility at many maturation-dependent chromatin regions.

The 2409 activity-regulated accessible regions constitute less than 4% of all dynamic in vitro peaks, but we noted that loss of activity has a broader effect on chromatin accessibility across many DIV28-specific accessible regions that do not reach statistical significance. Plotting maturation-dependent changes vs. TTX dependent changes in accessible peaks shows that loss of activity is negatively correlated with maturation between DIV7 and DIV28 ($R = -0.46$, *p* value <2.2e−16), but not between DIV0 and DIV7 ($R = -0.06$, *p* value <2.2e−16) (Supplementary Fig. 10c, d). We additionally observed an overall ~25% reduction in accessibility across the top 10k regions that gain accessibility between DIV7 and DIV28 in response to TTX treatment, but no such effect in DIV7 culture (Fig. 6e). These data show that loss of activity affects chromatin accessibility specifically during the later stages of maturation, consistent with in vivo motif-enrichment analysis (Fig. 3a, b) and the reported increase in synaptic activity in cultured motor neurons after DIV6[84].

Analysis of the RNA-seq data also showed that activity is required for the correct regulation of genes in the later stage of maturation. Differential gene expression[41,42] analysis identified 89 upregulated and 108 downregulated genes in DIV28 + TTX condition compared to DIV28 (*p* value <0.001, fold change >2) (Supplementary Fig. 10b). Plotting the expression levels of affected genes at DIV0, DIV7, and DIV28 shows that the expression of genes downregulated in the absence of activity normally increases from DIV7 to DIV28, while the converse is true for upregulated genes (Fig. 6f, g). The 197 genes that are significantly perturbed after the loss of activity only represent ~5% of all genes that change expression during maturation. However, similar to accessibility changes, we observe that loss of activity affects maturation-related gene expression changes more globally specifically between DIV7 and DIV28 (Fig. 6h). By plotting the expression changes of all in vivo and in vitro co-regulated genes against activity-dependent gene expression changes, we observe that changes in expression between DIV7 and DIV28, but not between DIV0 and DIV7, are negatively correlated with the presence of TTX (correlation: $R = -0.51$, *p* value <2.2e−16 for DIV7–DIV28; $R = 0.033$, *p* value = 0.11 for DIV0–DIV7) (Fig. 6h). We further validated the temporal specificity of the effects by quantifying the expression of *Thy1*, which is induced early during maturation, and *Spp1*, which is induced later (Fig. 4j, k). Whereas TTX has no effect on *Thy1*, it significantly reduces *Spp1* expression at the later stages of maturation (Fig. 6j).

Although we identified a negative correlation between loss of activity and maturation-dependent accessibility and expression changes, our data also show that most maturation-dependent transcriptional changes are not strongly affected by the loss of activity despite the commanding presence of AP-1 motifs in maturation-dependent

accessible regions. Indeed, PCA analysis of the accessibility and expression data confirmed that while DIV28 + TTX cells are shifted toward a more immature state, they still cluster closer to DIV28 (Fig. 6i). Together these data suggest that activity-dependent transcription factors function as modulators, rather than drivers of the neuronal maturation program.

## Discussion

Most postmitotic nerve cells are specified during embryonic development and maintained throughout life, undergoing functional changes as the nervous system matures and ages[1]. Great strides have been made in our understanding of cell-type specification during the transition from progenitor to postmitotic state, but the regulatory mechanisms that orchestrate changes in maturing postmitotic neurons remain unknown. This discrepancy is particularly apparent in in vitro studies—while diverse neuron types can be efficiently differentiated from stem cells, culture systems that recapitulate maturation remain scarce[16]. In this work, we have mapped the transcriptional profiles of in vivo mouse spinal motor neurons throughout life, uncovered putative regulators of maturation, and characterized a culture system that not only recapitulates aspects of in vivo maturation, but also allows functional interrogation of regulators.

Temporal profiling of motor neurons showed that gene expression and chromatin accessibility are dynamic for four weeks, from the time of motor neuron specification to the third postnatal week, and then remain remarkably stable for the rest of life. While some of the core motor neuron effector genes, including those that control cholinergic neurotransmission and motor pool identity, become induced soon after progenitor cells become postmitotic, other classes of effector genes, such as channels and receptors, are remarkably dynamic during maturation. This finding is consistent with previous studies that have shown that motor circuits and motor behaviors become adult-like only in the third postnatal week[6,19]. The finding that gene expression patterns remain stable between P21 and the end stage of life is also striking as it suggests that once an adult-like gene expression profile is acquired, aging and age-related diseases are likely to be primarily modulated by post-transcriptional mechanisms.

Interestingly, longitudinal gene expression studies performed on the whole cortex found that transcriptional identities stabilize around ~P30[85]. As cortical neurons are both regionally and functionally distinct from spinal motor neurons and a majority of them are born days after motor neurons in mice, these observations suggest a shared timeline of maturation for the entire nervous system. This led us to ask whether gene expression changes during maturation are neuron-type specific or largely universal. Comparative analysis revealed that despite the shared timeline, the motor neuron maturation program is largely distinct from cortical neurons, implying that the functional diversity of the adult nervous system continues to be refined during postnatal maturation.

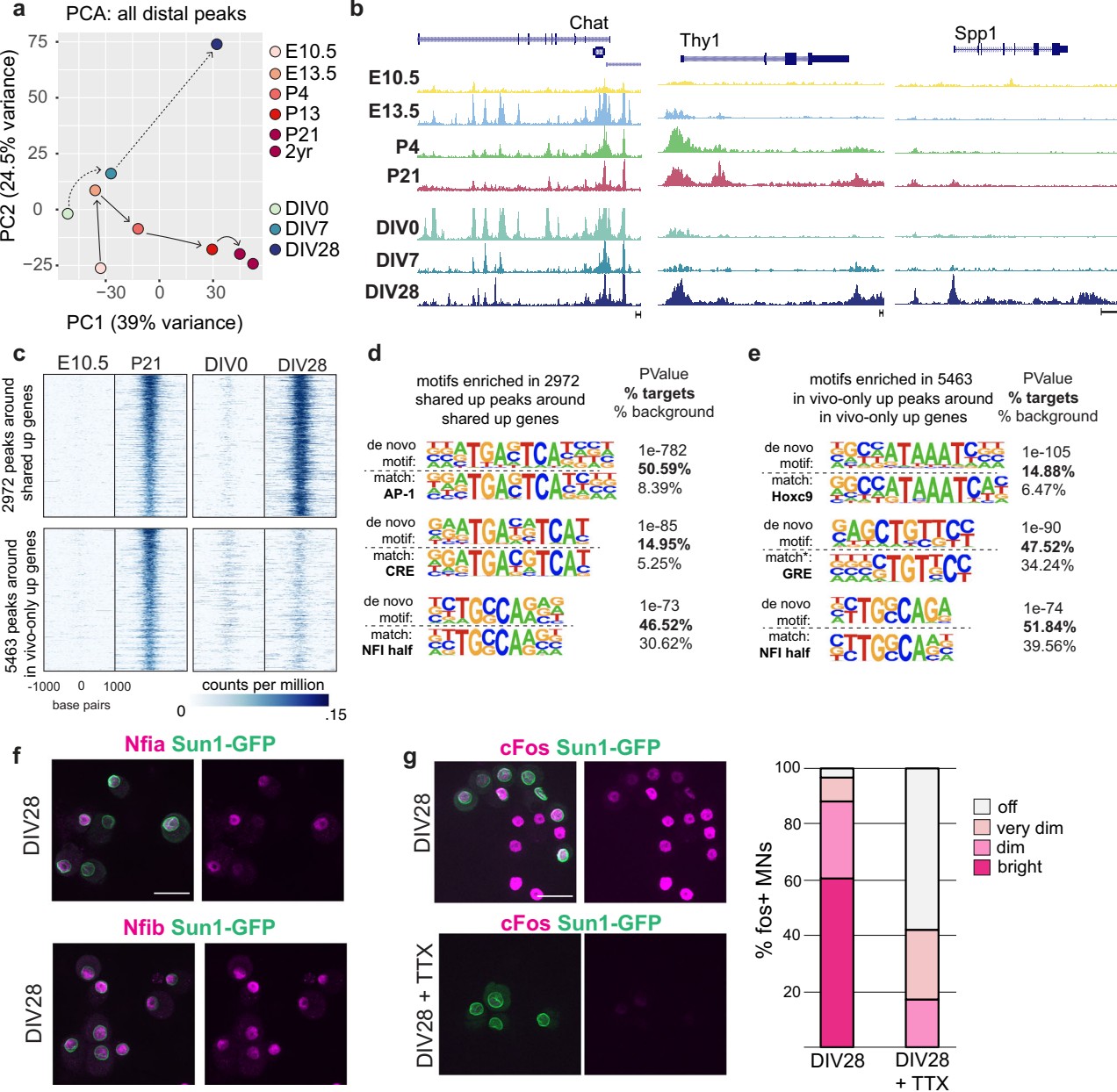

**Fig. 5 | Identification of shared and in vivo-specific regulators of maturation.**
**a** Principal component analysis on in vivo and in vitro datasets based on all distal ATAC-seq peaks. Each circle is the cumulative ATAC-seq data from two biological replicates. **b** ATAC-seq reads around a continuously expressed gene (*Chat*) and genes upregulated during maturation (*Spp1* and *Thy1*). **c** Heatmaps of ATAC-seq reads. Each panel spans ±1 kb from the center of peaks. Top: peaks upregulated both in vivo and in vitro around shared upregulated genes. Bottom: peaks upregulated only in vivo around genes upregulated only in vivo. **d**, **e** Motifs enriched in top (**d**) and bottom (**e**) genomic regions from (**c**). HOMER outputs show the de novo motif (top) and the best-matched known transcription factor motif (bottom) along with *p* value and prevalence. **f** Immunostaining of *Nfia* and *Nfib* in DIV28 cultures. **g** Immunostaining and scoring of *Fos* in DIV28 cultures in the absence and presence of TTX. All scale bars are 50 μm.

To identify potential regulators of maturation, we performed motif-enrichment analysis on age-specific accessible chromatin regions. This method proved to be highly effective, leading to the identification of a cascade of transcription factors that might control the maturation process. While it is well established that motor neuron specification is controlled by the LIM homeodomain transcription factors ISL1 and LHX3[22], our data show that embryonic maturation likely requires additional input from NFI factors, and that postnatal maturation continues to engage NFI, in conjunction with activity-dependent, and hormone receptor transcription factors.

The detailed transcriptional maps of in vivo neurons provided us the opportunity to identify the aspects of maturation that can be

recapitulated in vitro. Stem cell-derived motor neurons have greatly facilitated studies of cell fate specification, but their utility for studying maturation and modeling adult-onset diseases has remained limited[2]. We found that co-culturing motor neurons with glial cells for four weeks results in recapitulation of ~40% of upregulated expression changes during in vivo maturation, including changes in expression of functionally relevant effectors such as channels, receptors, synaptic, and structural genes. This result was surprising as motor neurons in culture are grown without the correct synaptic partners or muscle targets and in the absence of other extrinsic factors available in vivo. The identified set of genes therefore represent a core maturation program that is shared between in vivo and in vitro systems.

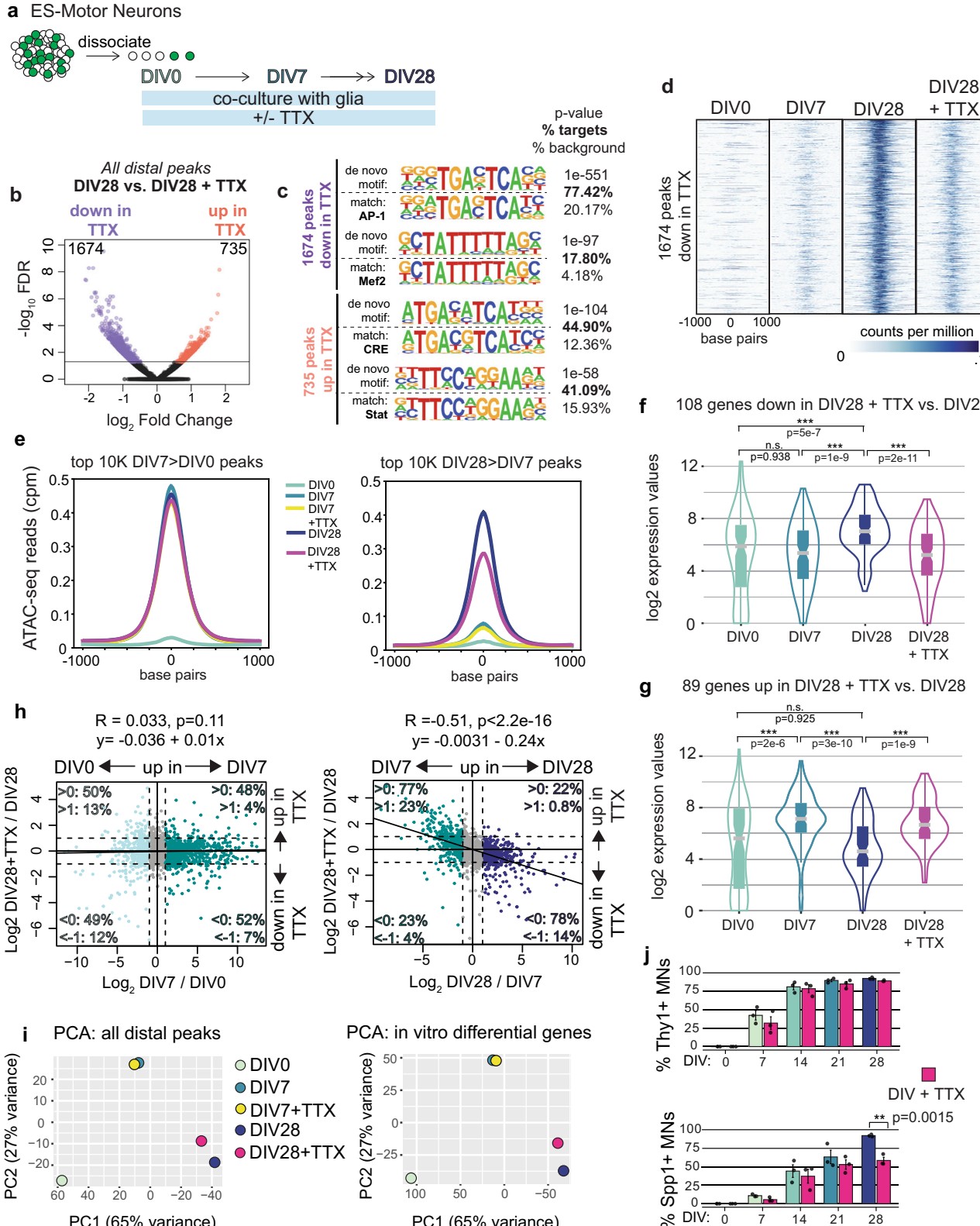

We found high enrichment of activity-dependent AP-1 motifs and moderate enrichment of NFI motifs in accessible chromatin regions around shared upregulated genes, allowing us to functionally test the role of neuronal activity in controlling transcriptional maturation. Sensory deprivation studies have long established the role of activity in the postnatal development of the cortex[14,15]. More recently, molecular studies have proposed that activity-regulated AP-1 factor FOS is the key

mediator of maturation in VIP and SST cortical neurons[9]. Our data showed that loss of activity only modestly perturbs the chromatin accessibility and gene expression program in cultured motor neurons, and that this effect is restricted to the later stages of maturation. In agreement with our findings, reduction of activity in vivo in SST inhibitory interneurons also showed a similar modest effect on gene expression during postnatal maturation[9]. These results allow us to

**Fig. 6 | Activity partially controls the later phase of in vitro maturation.**
**a** Schematic of loss of function experiments in the presence of TTX. **b** Differential accessibility between DIV28 and DIV28 + TTX conditions. Each dot represents a single ATAC-seq peak. Peaks in orange gain accessibility and peaks in purple lose accessibility in the absence of activity (FDR < 0.05, as calculated by DiffBind). **c** Top two motifs enriched in peaks that lose (top) or gain (bottom) accessibility in the absence of activity. HOMER outputs show the de novo motif (top) and the best-matched known transcription factor motif (bottom) along with $p$ value and prevalence. **d** Heatmaps show ATAC-seq reads at 1674 chromatin regions that lose accessibility in the absence of activity. Each panel spans ±1 kb from the center of the peak. **e** Lineplots showing cumulative accessibility at top 10k chromatin regions that gain accessibility between DIV0 and DIV7 (left) and between DIV7 and DIV28 (right). **f**, **g** In vitro expression of genes up- or downregulated in the absence of activity. ***$p$ value <1e−5 calculated by two-tailed $t$-tests; $n$ = 108 down and 89 up

genes. **h** Correlation between maturation and activity-dependent gene expression changes. Each dot on the plot represents a gene, the $x$-axis plots the change in expression over time, and $y$-axis plots the change in expression in TTX condition compared to no TTX at DIV28. Genes colored in shades of blue are up or down-regulated at those ages with $p < 0.001$ and fold change >2 (EdgeR, Fig. 4d). The percent of the colored genes in each quadrant with $\log_2$ fold change > 0 or 1 is also reported. **i** Principal component analysis of in vitro chromatin accessibility (left) and expression data (right). Each dot represents combined data from 2-3 replicates. **j** Scoring of Thy1-P2A-SUN1-GFP reporter and SPP1 protein in cultured motor neurons in the presence or absence of activity, as detected by immunostaining. Error bars show SEM; $n$ = 3 independent biological replicates shown as black dots. For box plots, center line is the median, the interquartile range is 25–75th percentile, and outliers are eliminated; $p$ values from two-tailed $t$-tests. Source data are provided as a Source Data file.

make two important conclusions. First, the in vitro system recapitulates in vivo-like temporal transitions from early to late regulators of maturation. Activity-dependent motifs were specifically enriched at later stages of maturation in vivo, and loss of activity only affects later stages of maturation in vitro. Second, neuronal activity is not the primary driver of the core maturation program, but rather functions as a modulator of the gene expression program by increasing chromatin accessibility and enhancing expression levels of many maturation genes.

Based on motif-enrichment and gene expression analysis in this study, both NFI factors and nuclear hormone receptors are promising activity-independent regulators of maturation. The NFI motif was predictive of global accessibility at all ages after E10.5, and it was highly enriched in new peaks that became accessible at consecutive ages, suggesting that NFI factors change their binding pattern over time. In immature E13.5 neurons, NFI-containing peaks were co-enriched for HOX and LIM motifs while in mature P21 neurons, they were co-enriched for activity-dependent AP-1 motifs. These facts together argue that NFI factors do not act as pioneer factors by themselves, but rather operate as cofactors that regulate chromatin accessibility in conjunction with other transcription factors. The potential involvement of steroid hormone receptors in maturation is also intriguing as it suggests a link between increased levels of circulating hormones in juvenile animals and neuronal maturation. While hormone receptor motifs were highly enriched in vivo, the absence of these motifs in vitro suggests that the maturation state of cultured neurons might be further enhanced by expression of relevant receptors and inclusion of their ligands in culture media.

Lastly, motor neurons exhibit notable changes in their susceptibility to degeneration during their maturation and adult life. Specifically, motor neuron survival is dependent on muscle-derived neurotrophic signals during a brief embryonic window of programmed cell death; at perinatal stages, motor neurons show selective sensitivity to decreased levels of SMN protein in mouse models of spinal muscular atrophy; and in adult animals carrying ALS causing mutations, there is a wide range of temporal onsets of motor neuron degeneration[56]. The transcriptional and chromatin characterization of motor neuron maturation in vivo, together with the identification of the core maturation program recapitulated in cultured neurons in vitro, provide a foundation for the development of more reliable models for late-onset neurodegenerative diseases.

## Methods
### Mice
Animals were handled according to protocols approved by the Institutional Animal Care and Use Committee at the Columbia Institute of Comparative Medicine. The animal care and use program at Columbia University is accredited by the AAALAC International and maintains an Animal Welfare Assurance with the Public Health Service, Assurance number D16-00003 (A3007-01). Mice were housed in a pathogen-free

barrier facility with a 12-h light/dark cycle. All mouse lines used have been previously characterized: SUN1-sfGFP-Myc[26] (JAX stock#021039), ChAT-IRES-Cre::SV40pA::Δneo (Chat-Cre)[25] (JAX stock#031661), and Hb9-GFP transgenic mice[24]. For purification of whole cells at E10.5, heterozygous Hb9-GFP males were crossed to wildtype C57BL/6J females, and heterozygous Hb9-GFP embryos were sacrificed for motor neuron purification. Three to five embryos were used per replicate for these experiments. For motor neuron nuclei purification at all other ages, homozygous SUN1-2xsfGFP-6xMYC mice were crossed to homozygous ChAT-IRES-Cre::SV40pA::Δneo mice and trans-heterozygous progeny were sacrificed for experiments. For each replicate of RNA-seq and ATAC-seq experiment at each age, spinal cord tissue was combined from four to seven animals. Both males and females were included in each experiment. Three replicates of RNA-seq experiments were performed at E10.5, P4, P21, and P56, and two replicates were performed at E13.5, P13, and 2 years. Two replicates of ATAC-seq were performed at all ages.

### Cell culture
MN media: neurobasal with 1:50 B27, 1:100 Glutamax, 100 µl 1:100 Beta-mercaptoethanol, 1:50 FBS, 1:100 Pen/Strep. For long-term co-cultures supplemented with GDNF (10 ng/ml, R&D systems 212-GD/CF), BDNF (10 ng/ml, Peprotech 450-02-50UG), CNTF (10 ng/ml, R&D Systems 257-NT/CF), IGF (10 ng/ml, R&D Systems 291-G1-200), 1:5000 UFDU for 1.5 weeks (10 mM each Uridine [Sigma-Aldrich; Catalog Number U3750] and Fluorodeoxyuridine [Sigma-Aldrich; Catalog Number F0503]).

Activity experiments: to block activity 1 µM of TTX was added to the MN media from DIV0 to DIV28.

Deriving primary astrocytes: astrocyte-enriched primary cortical glial cultures were acquired from P2 mouse cortices as previously described[86]. Plates were shaken in 37 °C for 6 h to reduce microglial contamination. Plated astrocytes were passaged once and then frozen in 90% FBS + 10% DMSO.

Deriving Chat-CRE;SUN1-GFP ESC line: Chat-Cre homozygous males were bred to SUN1-GFP homozygous female, and trans-heterozygous blastocysts were collected for stem cell derivation as previously published[79].

**Preparation of co-culture plates.** Coating plates: 6-cm or 96-well plates were coated with poly-ornithine and laminin. Plates were incubated in poly-ornithine (1:5 in H2O, Sigma-Aldrich P4957-50ML) for 3 h or overnight at 37 °C, washed 3× with PBS to fully remove poly-ornithine, and incubated with Laminin (1 mg/ml, 1:250 in ice-cold PBS) for 3 h or overnight at 37 °C.

Glial cultures: astrocytes were thawed on T75/T175 flasks (~6000 cells/cm²) coated with 0.1% gelatin, and allowed to grow to confluence (7–9 days). Cells were dissociated with 0.05% trypsin, passaged 1:1 (surface area), and replated on Poly-ornithine/Laminin coated 6-cm or 96-well plates. Cells were allowed to grow to confluence for 5–7 days,

and dissociated motor neurons were plated on top. Astrocytes contact inhibit once they are confluent and are not affected by UFDU in motor neuron media.

Motor neurons: embryonic stem cells were differentiated into motor neurons following published protocols[80]. On day 6 of differentiation, embryoid bodies (EBs) were dissociated with Accumax as follows: EBs from each 10 cm plate were pelleted and resuspended in 1 ml Accumax with 1:200 5 mg/ml DNAseI, incubated in a 37 °C water bath for 10 min, and triturated with a 1 ml pipette ~30× times till a cloudy cell suspension was visible (instead of intact EBs). Then, 1 ml of warm MN media was added to Accumax cell suspension, and cells were spun down at 300 g for 5 min, resuspended in warm motor neuron media, filtered through at 40 µm cell strainer, and counted and plated on astrocytes. A total of 12,000 dissociated cells were plated on each well of 96-well plates and 2 million dissociated cells were plated on 6 cm plates in motor neuron media. In all, ½ media was changed every 3 days. A large number of cells do not survive the plating.

96-well plates were used for imaging-based experiments. Six-cm plates were used for RNA-seq and ATAC-seq experiments.

### CRISPR-Cas9-mediated Thy1 reporter
WT Black6 embryonic stem cells were transfected with a gRNA targeted close to the stop codon of Thy1 gene (GACTTCATTTCTCTGT-GAC), a repair template (500 bp homology arm–P2A-SUN1-GFP-3xMyc −440 bp homology arm), and a plasmid containing Cas9-mcherry. mCherry positive cells were FAC sorted ~48 h later and grown at low density (6000 cells in a 6-cm plate with MEFs) till colonies were visible (~7–8 days). Single colonies were picked, dissociated, and clonally plated on two 96-well plates. One 96-well plate was genotyped using internal and external primers and positive clones were expanded from the second 96-well plate.

### Immunostaining
E10: brachial, thoracic, lumbar vertebral columns were dissected from embryos and fixed in 4% paraformaldehyde for 2 h. Fixed vertebral columns were washed in PBS 3× times for 10 min, 30 min, and 1 h. Spinal cords were carefully dissected out of vertebral columns and cryopreserved in OCT and sectioned with a cryostat into 15 µm sections on slides.

Sections were blocked for 30 min in Blocking Buffer (10% donkey serum (EMD Millipore), 0.2% Triton X-100 (Sigma-Aldrich) in 1X PBS, and 0.05% NaN3), followed by primary antibody treatment for 1hr at room temperature or overnight at 4 °C in Antibody Buffer (2% donkey serum, 0.2% Triton X-100 in 1X PBS and 0.05% NaN3), followed by 3× 5-min washes with Wash Buffer (0.1% Triton X-100 in PBS), followed by secondary antibody treatment in Antibody Buffer for 2 h at room temperature, followed by 1× wash with Wash Buffer + DAPI for 15 min, and 2× washes with just Wash Buffer for 5 min each. Sections were washed one last time with PBS, and sealed with Flouromount G (Thermo Fisher Scientific OB100-01) and coverslips.

P4, P21, and P56: deeply anesthetized mice were transcardially perfused with 20 ml of PBS, followed by 20 ml of 4% paraformaldehyde. Vertebral columns were dissected out and fixed for 30 min or overnight in 4% paraformaldehyde at 4 °C, followed by 3 PBS washes lasting 1 and 2 h, and overnight. Brachial, thoracic, lumbar spinal cords were dissected out of the vertebral columns and sectioned with a vibratome in 30–70 µm sections. Immunostaining was performed in free-floating sections. Sections were incubated in primary antibody for 48 h at room temperature in Antibody Buffer (1% BSA, 0.1% Triton X-100 in 1X PBS, and 0.05% NaN3), followed by 3× 1–3 h washes with Wash Buffer (0.1% Triton X-100 in PBS), followed by secondary antibody treatment in Antibody Buffer overnight at room temperature, followed by 1× wash with Wash Buffer + DAPI for 1 h, and 2× washes with just Wash Buffer for 30 min to 1 h each. Spinal cords were placed on slides and sealed with Flouromount G (Thermo Fisher Scientific OB100-01) and coverslips.

Cell cultures and embryoid bodies: EBs were fixed for 10 min in 4% PFA on ice, cryopreserved as previously described (Wichterle and Peljto, 2008), and sectioned into 10 µm slices on slides using a cryostat. Motor neuron co-cultures were fixed for 10 min in 4% PFA on ice, and stained in 96-well plates. Cells were incubated in primary antibody for 1–2 h at room temperature in Antibody Buffer (1% BSA, 0.1% Triton X-100 in 1X PBS, and 0.05% NaN3), followed by 3× 5-min washes with Wash Buffer (0.1% Triton X-100 in PBS), followed by secondary antibody treatment in Antibody Buffer for 45 min at room temperature, followed by 1× wash with Wash Buffer + DAPI for 15 min, and 2× washes with just Wash Buffer for 5 min each. EB sections were sealed with Flouromount G (Thermo Fisher Scientific OB100-01) and coverslips. Cultures in 96-well plates were covered with 150 µl of PBS and the plate was sealed with parafilm.

The following primary antibodies were used: GFP (Chick, Thermo Fisher A10262, 1:3000), Chat (Goat, EMD Millipore AB144P 1:100), hb9 (Guinea pig 1:100 from Jessell Lab), NeuN (Rabbit, Millipore-Sigma ABN78, 1:1000), Spp1 (Mouse, R&D Systems AF808, 1:50; Goat, sc-21742, 1:300), Isl1/2 (Ms 4D5 DSHB optimized concentration by staining; Neuromics GT15051 1:5000); Nfia (Rb, Active Motif 39397, 1:1000), Nfib (Rb, Active Motif 39091, 1:1000), NFH (Chk, Neuromics CH22104, 1:2000), cFos (Rat, Synaptic Systems 226 017, 1:500), Nr3c1 (Rabbit, Invitrogen PA1-511A, 1:2000), Nr3c2 (DSHB clones 1D5 and 3F10, 1:33). The following secondary antibodies were used: 1:800, 706-165-148, 711-165-152, 715-165-151, 715-545-150, 712-605-153, 712-605-152, 712-605-151, 703-545-155, 715-175-151, 715-175-150 from Jackson ImmunoResearch Laboratories.

### Imaging
Images were acquired either with 20×, 40× oil, 60× oil objectives using confocal laser scanning microscope (LSM Zeiss Meta 510 or 780) or with an apotome on an inverted phase contrast fluorescence microscope (Zeiss Observer. Z1). Images were processed using Zen and ImageJ.

### Purification of whole motor neurons from Hb9-GFP mice for RNA-seq and ATAC-seq
Brachial, thoracic, lumbar spinal cords were dissected out of E10.5 embryos in HBSS buffer. Spinal cords were cut into small pieces, pipetted into 1.5 ml tubes, and pelleted with a brief spin. Spinal cord tissue was then resuspended in Accumax for dissociation. 1 ml of Accumax (Sigma-Aldrich A7089-100ML) with 5 µl of 5 mg/ml DNAseI (Millipore-Sigma DN25-10MG) was used for two spinal cords. Dissociation was performed by incubating tubes at 37 °C while shaking at 1000 rpm for 15 min. Tissue was then homogenized by pipetting up and down ~20 times, and mixed with 3 ml of L15 media (with 1:200, 5 mg/ml DNAseI). The resulting cell suspension was spun down at 300 g for 5 min to pellet single cells. Cells were resuspended in Sorting Buffer (Ca/Mg free PBS with 1% FBS), and filtered through a 20 µm filter. High GFP cells were sorted using a Bio-Rad Se3 sorter into Sorting Buffer. For RNA collection, cells were spun down, resuspended in TRIzol (Life Technologies 5596-018), flash frozen, and stored at −80 °C till RNA purification. For ATAC-seq, cells were spun down and ATAC-seq was performed exactly as published in ref. [59].

### Modified INTACT nuclear isolation for RNA-seq
We used a modified version of the INTACT[26] protocol with all the same chemical reagents. The following volumes are for collecting nuclei from three to four spinal cords from E13.5/P4 mice or two spinal cords from P13 or older mice. Volumes have to be adjusted for higher numbers of spinal cords being processed together. Brachial and lumbar regions of the spinal cords were dissected and processed together

for RNA-seq. Brachial, thoracic, and lumbar regions of the spinal cords were dissected and processed together for ATAC-seq.

Purification of nuclei: spinal cords were homogenized by douncing 10× times (E13.5, P4) or 15–20× times (P13, P21, P56, 2 years) with a 1 ml dounce-homogenizer in a total of 5 ml Buffer HB (0.25 M sucrose, 25 mM KCl, 5 mM MgCl2, 20 mM Tricine-KOH pH 7.8, 1 mM DTT, 0.15 mM spermine, 0.5 mM spermidine, Roche protease inhibitor tablets) supplemented with 1.5 µl/ml RNAse Inhibitor (RNasin Plus RNase Inhibitor, Promega N2615) and 0.3% IGEPAL CA-630. Homogenized spinal cords were filtered through a 100-µm strainer and combined with 1:1 volume of 50% iodixanol (5 volume of OptiPrep[60% iodixanol] + 1 volume of Diluent [150 mM KCl, 30 mM MgCl2, 120 mM Tricine-KOH pH 7.8]), creating a 10 ml suspension that is 25% iodixanol. The 25% iodixanol suspension was split equally into two 15 ml falcon tubes, such that each tube had 5 ml. The 5 ml 25% iodixanol suspension in each 15 ml tube was underlaid with 2 ml of 40% iodixanol (50% iodixanol solution diluted in Buffer HB). Tubes were spun at 1000 g for 12 min at 4 °C with slow acceleration and deceleration (set to "3"). After the spin, a layer of cellular debris collects on top of the 25% suspension. This was cleared off by pipetting. 1.2 ml of nuclei were then pipetted from the interface of 25% and 40% layers from each 15 ml tube. The nuclei were dounce-homogenized 5× times in a 1 ml dounce-homogenizer. This step is necessary for eliminating most of the clumps that form when nuclei stick together during the spin, and increases the final purity of Sun1-GFP+ nuclei.

Affinity purification of GFP+ nuclei: 1.2 ml of nuclei were pipetted into a 1.5 ml tube. 70 µl of ProteinG Dynabeads (Life Technologies, 100004D) were washed 2× times with 800 µl wash buffer (0.25 M sucrose, 25 mM KCl, 5 mM MgCl2, 20 mM Tricine-KOH pH 7.8, 0.4% IGEPAL CA-630, 1 mM DTT, 0.15 mM spermine, 0.5 mM spermidine) and returned to their original volume (70 µl) in wash buffer. 10 µl of washed ProteinG beads were added to 1.2 ml of nuclei for preclearing, for which tubes were incubated at 4 °C for 30 min with rotation. At this point, the material bound to the beads is non-specific and needs to be removed from solution. To do this, tubes were placed on a magnet for 2 min and 1.2 ml of precleared nuclear suspension was then pipetted into a new 1.5 ml tube (leaving behind preclearing beads, which can be discarded). For IP reaction, the following was then added to each tube containing 1.2 ml of nuclei: 5ul anti-GFP antibody (Thermo Fisher Scientific G10362), 200 µl Wash Buffer, 1 µl RNase Inhibitor. The IP reactions were incubated for 30 min at 4 °C with rotation. After incubation, 25 µl pre-washed beads were added to each IP reaction, followed by an additional 20 min of rotation at 4 °C. At this point the samples contain beads that are bound to GFP+ nuclei. To increase the number of beads bound to each nucleus the following was repeated 6–7 times: tubes were placed on a magnet for no more than 1 min, then put on ice for ~15 s, then beads were resuspended into solution by gentle inversion. The tubes were then put on a magnet for 2 min and the liquid suspension (which contains mainly GFP-nulcei) was pipetted off. The beads from all tubes that came from the same starting sample were combined and resuspended in 700 µl of wash buffer and filtered through a 20 µm filter to eliminate large clumps of nuclei. The beads were then washed 5× times by doing the following: tubes were placed on a magnet for 1 min, Wash Buffer was pipetted off, 1.4 ml of fresh Wash Buffer was added, tubes are placed on ice for ~15 s, and beads were resuspended by rigorously pipetting. After the last wash, beads were resuspended in 200–300 µl of Wash Buffer. 1.5 µl of beads were mixed with 1.5 µl of DAPI for visualization with a fluorescent microscope and determination of purity. Finally, the tubes were placed on a magnet, Wash Buffer was removed, beads were resuspended in 500 µl of Trizol, and incubated at room temperature for 10 min. The tube was then placed on the magnet for the last time, 500 µl of TRIzol was pipetted into a new tube that was flash frozen and stored at −80 °C till RNA purification. This procedure yielded ~10,000 nuclei per spinal cord (using brachial and lumbar sections). At E13.5 and P4, nuclei were ~85% GFP+ with some contamination from blood vessels. At P13 and older ages, nuclei were >95% GFP+.

## FAC sorting nuclei for ATAC-seq

We noticed that bead-bound nuclei purified using the above affinity purification method yielded a very low concentration of DNA after ATAC-seq. We therefore FAC sorted nuclei for ATAC-seq experiments. To do this, nuclei were purified from spinal cords following the above method. Then, instead of performing affinity purification, we did the following:

FAC sorting of GFP+ nuclei: 1.2 ml of nuclei were thoroughly mixed with 3.6 ml of Wash Buffer in 5 ml Polypropylene Round-Bottom Tube. Tubes were spun at 500 g for 5 min to pellet nuclei. The liquid suspension was carefully pipetted off. All nuclei from the same starting sample were combined and resuspended in 1.5–2.5 ml of Wash Buffer with rigorous pipetting to ensure no clumps of nuclei remained. The nuclei were then filtered through a 20-µm filter, and 1.5 µl/ml of RNAse inhibitor was added. GFP+ nuclei were then FACs sorted into Wash Buffer with 1.5 µl/ml of RNAse inhibitor using a Bio-Rad Se3 sorter. 1.5 µl of sorted nuclei were mixed with 1.5 µl of DAPI to visualize nuclei and determine purity. This procedure resulted in >95% pure nuclei at all ages.

## Modified INTACT purification of motor neuron nuclei from embryoid bodies and co-cultures for RNA-seq

Releasing nulcei from embryoid bodies: EBs from each 10 cm plate were pelleted and resuspended in 2 ml cold lysis buffer (10 mM Tris-HCl pH 7.4, 10 mM NaCl, 3 mM MgCl2, 0.1% IGEPAL CA-630) in 15 ml falcon tube and incubated on ice for 10 min followed by trituration (30×) with 1 ml pipette, followed by an additional 10 min incubation on ice and a second 30× trituration. 1ul of the nuclei mix was stained with DAPI and observed under fluorescence to ensure that cells had lysed.

Releasing nuclei from co-cultures: Media was removed from 6 cm plates and 1 ml of cold lysis buffer was added (10 mM Tris-HCl pH 7.4, 10 mM NaCl, 3 mM MgCl2, 0.1% IGEPAL CA-630). Cells were scrapped with a cell scraper, and 1 ml of media + cells were placed in 15 ml tube on ice. All plates of the same genotype for each experiment were mixed together. Buffer + cells were triturated 30× times with 1 ml pipette followed by a 10 min incubation of ice followed by another 30× trituration. 1 µl of the nuclei mix was stained with DAPI and observed under fluorescence to ensure that cells had lysed.

Purification of nuclei: nuclei in lysis buffer were spun at 4 °C in a hanging centrifuge at 500 g for 5 min. For each 15 ml tube, lysis buffer was pipetted off and nuclei were resuspended in 2 ml Buffer HB (0.25 M sucrose, 25 mM KCl, 5 mM MgCl2, 20 mM Tricine-KOH pH 7.8, 1 mM DTT, 0.15 mM spermine, 0.5 mM spermidine, Roche protease inhibitor tablets) supplemented with 1.5 µl/ml RNAse Inhibitor (RNasin Plus RNase Inhibitor, Promega N2615) and 0.3% IGEPAL CA-630. 2 ml of 50% iodixanol (5 volume of OptiPrep[60% iodixanol] + 1 volume of Diluent [150 mM KCl., 30 mM MgCl2, 120 mM Tricine-KOH pH 7.8]), was added to this, creating a 4 ml suspension that is 25% iodixanol. The 4 ml 25% iodixanol suspension in each 15 ml tube was underlaid with 2 ml of 40% iodixanol (50% iodixanol solution diluted in Buffer HB). Tubes were spun at 1000 g for 12 min at 4 C with slow acceleration and deceleration (set to "3"). 1.2 ml of nuclei were then pipetted from the interface of 25% and 40% layers from each 15 ml tube. The nuclei were dounce-homogenized 10× times in a 1 ml dounce-homogenizer. This step is necessary for eliminating most of the clumps that form when nuclei stick together during the spin, and increases the final purity of Sun1-GFP+ nuclei.

Affinity purification of GFP+ nuclei: same as for spinal cords.

## FACs of motor neuron nuclei from embryoid bodies and co-cultures for ATAC-seq

Nuclei from EBs and co-cultures were purified as above, and then FACs were sorted the same as spinal cord nuclei.

## RNA purification

Tubes containing samples in TRIzol were thawed to room temperature, 100 μl of Chloroform was added to 500 μl of TRIzol, mixed by rigorous inversion, and spun down at 16,000 g for 15 min. ~270 μl of the aqueous layer was carefully removed. RNA was purified from the aqueous layer using the Zymo RNA microprep kit (ZymoResearch, cat #R2060) following the manufacturer's instructions.

## RNA-seq library preparation and sequencing

RNA-seq libraries were prepared at the MIT BioMicroCenter using the following kit: Clontech SMARTer Stranded Total RNA-Seq Kit−Pico Input Mammalian with ZAPr ribosomal depletion. Paired-end sequencing was performed using the 150nt Nextseq kit.

## RNA-seq processing

Reads were trimmed for adaptors and low-quality positions using Trimgalore (Cutadapt v0.6.2)[87]. Reads were aligned to the mouse genome (mm10) and gene-level counts were quantified using RSEM (v1.3.0)[88] rsem-calculate-expression using default parameters and STAR (v2.5.2b) for alignment. Read counts were normalized by the median of ratios normalization[89] using a Python (v3.6.9) script. All reported expression values are normalized expression values. For Figs. 1−3 and Supplementary Figs. 2, 3, 4, 5b−e, and 6 all motor neuron in vivo data were normalized together. For Supplementary Fig. 5a P4, P56 motor neuron data were normalized with P7, P56 cortical data. For Fig. 4 and Supplementary Fig. 8, in vivo and in vitro (no TTX) motor neurons data were normalized together. For Fig. 6, all in vitro data (including TTX at DIV28) were normalized together. All normalized expression values are provided in Supplementary Tables 1 and 3.

All reported differential gene expression analysis was performed on RSEM gene-level read counts using EdgeR[41,42]. For each set of comparisons being performed, genes were secondarily filtered for those that are expressed with CPM ≥5 in all replicates in at least one condition. For example, for Fig. 1, all datasets from E10.5, E13.5, P4, P13, P21, P56, and P2yr were loaded to EdgeR, and filtered for genes that are expressed with CPM ≥5 in all replicates at any age. Differential gene expression was then performed between subsequent ages.

## Selecting effector genes for Fig. 1h

Genes belonging to functional categories were selected on the basis of their Refseq descriptions or gene names. Channels are all genes that include the word "channel" in their description. Receptors are all genes that include the word "receptor" but not the word "nuclear" in their description. Collagens are all genes that begin with "Col". Transcription factors are all genes that include "transcription" or receptors that include the word "nuclear" in their description. Chromatin regulators are all genes that include "histone" or "chromatin" in their description. Kinases are genes that include "kinase" in their description. All genes that are expressed with CPM ≥5 in all replicates at any in vivo age are considered expressed in motor neurons. Of these genes, we calculated the percent that had <100 average normalized expression value at E10.5, but increased 2-fold (p value <0.001, as determined by EdgeR) between E10.5 and P21. Expression values of these genes are provided in Supplementary Table 2.

## ATAC-seq on nuclei

FAC sorted nuclei were spun at 500 g for 7 min to pellet nuclei. Wash Buffer was carefully pipetted off, leaving behind ~100 μl of Wash Buffer. The nuclear pellet was gently resuspended in the remaining 100 μl Wash Buffer and transferred to a 0.2 ml PCR tube. Nuclei were pelleted again by spinning at 500 g for 5 min. Almost all of the Wash Buffer was then pipetted off. Nuclei were then resuspended in the tagmentation solution containing Tn5. ATAC-seq was performed using the OMNI-ATAC protocol[62]. The cell lysis step was skipped as we were starting with nuclei, but the remainder of the protocol was followed exactly as published.

## ATAC-seq library preparation and sequencing

ATAC libraries were amplified by following the published protocol[59]. All ATAC-seq libraries required eight to nine total amplification cycles. ATAC-seq libraries were sequenced at the MIT BioMicroCenter. Paired-end sequencing was performed using the 75nt Nextseq kit.

## ATAC-seq processing

Reads were trimmed for adaptors and low-quality positions using Trimgalore (Cutadapt v0.6.2)[87]. Reads were aligned to the mouse genome (mm10) with bwa mem (v0.7.1.7)[90] with default parameters. Duplicates were removed with samtools (v1.7.2)[91] markdup and properly paired mapped reads were filtered. Accessible regions were called using MACS2 (v2.2.7.1)[92] with the parameters -f BAMPE -g mm -p 0.01−shift −36−extsize 73−nomodel−keep-dup all−call-summits. Accessible regions that overlapped genome blacklist regions were excluded from downstream analysis. The BEDTools[93] suite was used to compare peaks between different samples. DiffBind[83] was used for differential accessibility analysis between DIV28 and DIV28 + TTX. For all analyses, except DiffBind, peaks called on pooled replicate data were used. For DiffBind replicate data were input independently.

Example of how percent ATAC peaks that change between subsequent ages were identified for Fig. 2b, and Supplementary Fig. 9b: to find peaks that are upregulated between E10.5 and E13.5, we took the top 100k peaks at E13.5 and subtracted all peaks identified in E10.5 using BEDTools. To find peaks downregulated between E10.5 and E13.5, we took the top 100k peaks at E10.5 and subtracted all peaks identified in E13.5 using BEDTools. All comparisons were performed in this way.

For calculating percent proximal and distal peaks in Fig. 2c and Supplementary Fig. 9d: For the left bar, the percent of proximal or distal peaks in the top 100k peaks at each age were calculated and averaged. For the right bar, the percent of proximal or distal peaks in ATAC-seq peaks that are differentially accessible at consecutive ages were calculated and averaged.

## Non-negative least squares deconvolution of bulk P56 data

Cervical, thoracic and lumbar P56 (male and female combined) single-cell RNA-seq datasets from Alkaslasi et al., 2021, were integrated using Seurat v4.0.6, following the selection of high-quality single-cell profiles (at least 1000 unique genes detected per cell). Integrated data was subsequently clustered using Seurat, which revealed multiple excitatory (Slc17a6+), inhibitory (Gad1+ and/or Gad2+) and cholinergic (Chat+) cell clusters. We then sampled the same number of cells from all clusters to avoid biasing against smaller clusters. The number of cells sampled was equal to the number of cells in the smallest cluster. We then obtained the mean log gene expression profile for each cluster and then based on Pearson correlation levels between pairs of clusters (calculated using top 1000 highest variance genes), we identified four transcriptionally distinct classes of cell types, each of which consisted of multiple, highly correlated clusters. The bulk p56 data were then deconvolved into a linear sum of the mean gene expression profiles of these four cell types using non-negative least squares fitting, which assigned all non-cholinergic cell types a weight of zero. Subsequently, we repeated this process with only cholinergic clusters to deconvolve the bulk data into classes of cholinergic cell types.

## Motif discovery analysis

HOMER[67] was used to perform de novo motif enrichment using the command findMotifsGenome.pl with the parameter size given. The de novo output was manually curated for instances when very similar motifs were found enriched twice in the same condition, only one was reported. Selection of genomic regions for HOMER analysis: we first filtered out genomic regions that are accessible in ESCs or bound by CTCF, a hallmark of sites controlling genome architecture[94–96], and then selected the top 10k peaks that are most significant at each age, or that are differential between consecutive ages. For differential motif activity, we trained a DeepAccess[65,66] model on 3,555,674 100nt regions. Each region is labeled as accessible or inaccessible in each of the eight cell types: ESCs, E10.5, E13.5, P4, P13, P21, P56, and P2yr. We define a region as accessible in a given cell type if more than 50% of the 100nt region overlaps a MACS2 accessible region from that cell type. 2,555,674 regions were open in at least 1 cell type, and 1,000,000 regions were closed in all cell types (randomly sampled from the genome). Chromosomes 18 and 19 are held out for validation and testing. Methods for computing differential expected pattern effects between cell types are described in[65]. Briefly, we compute a differential expected pattern effect as the ratio between the effect that the presence of a transcription factor motif has on the predicted accessibility of a DNA sequence in one cell type relative to another cell type within a DeepAccess model. We use a consensus database of 108 transcription factor motifs representing the major transcription factor families that we derived from the HOCOMOCOv11 database.

## Pathway enrichment analysis

The web versions of Reactome[43] (https://reactome.org/PathwayBrowser/#/) and SynGo[81] (https://www.syngoportal.org/index.html) were used for pathway enrichment analyses.

## Conservation analysis

Positional phastCons and PhyloP mouse conservation scores[97] were downloaded as bigwigs from UCSC. Average per-base conservation scores over bed-accessible regions were calculated using a Python (v3.6.9) script. For comparison, we compare conservation within accessible regions to the conservation of 49,896 2 kb regions randomly sampled from the genome.

## PCA

Using a python script, normalized gene counts are log-transformed and filtered to keep only genes with more than 10 normalized reads in at least 1 sample. The gene by sample matrix is mean-centered and scaled to unit variance and used as input to perform PCA. Scaling and PCA was performed in Python (v3.6.9) with the sklearn package.

## Published data used in this study

The cortical RNA-seq and ATAC-seq data from Stroud et al., 2020, was downloaded from Gene Expression Omnibus (GEO): GSE150538.

The ESC ATAC-seq data from Dieuleveult et al., 2016, was downloaded from GEO: GSE64825.

Raw data from cortical neurons and ESCs were downloaded and processed as described above.

The snRNA-seq data from Blum et al., 2021, are on GEO: GSE161621. The snRNA-seq data from Alkaslasi et al., 2021, are on GEO: GSE167597. Integrated snRNA-seq data from Blum et al., 2021, and Alkaslasi et al., 2021, are from spinalcordatlas.org. A list of alpha- and gamma-specific genes was integrated from Blum et al., 2021, and Alkaslasi et al., 2021. A list of pool identity markers was taken from Blum et al., Fig. 4c. Effectors genes for cholinergic cell types in Supplementary Fig. 2g were integrated from Blum et al., 2021, and Alkaslasi et al., 2021 (MNgene: *Tns1, Gm43122, Zbtb16, Pdgfd, Anxa4, Plch1, Esrrb, Htra1, Plekhg1, Pald, Bcl6, Ret, 6030407O03Rik, Gm13912, Aox1,*

*Glis3, Grin3b, Esrrb, Rreb1, Chst9, Ahnak2*; VMgene: *Fbn2, Fign, 9530026P05Rik, Mme, Gm29683, Cpa6, Col8a1, Fam19a2, Hpgds, Nox4, Cpne4, Qrfpr, Pard3b, Trabd2p, Rxfp1, Pde8a, Vwa5b1, Qrfpr, Gnb4*; INgene: *Mpped2, Grik3, Dscam1l, Slc6a1, Nxph2, Pou6f2, Il1rapl2, Gm20754, Synpr, Arhgap6, Kcnmb2, Satb1, Pde11a, C1qtnf7, Bcl11a, Gm12649*).

## Unique biological materials

New cell lines generated in this study will be made available upon reasonable request.

## Statistics and reproducibility

The algorithms used for peak calling for ATAC-data, motif-enrichment (HOMER and DeepAccess), differential gene expression (EdgeR), differential accessibility (DiffBind), regression analysis (using linear model (lm) in R), and pathway enrichment (Reactome, SynGo) provide $p$ values, $q$ values, and/or FDR. Computation was performed using R, Phython, or MatLab. Methods used to generate $p$ values for comparison of expression data are noted in figure Legends. All immunostainings shown were performed on at least three mice (mix of male and female). Quantification of transcription factor staining in Fig. 3 was performed by scoring at least three sections from each of the three mice at E13.5, P4, and P56. For quantification of *Spp1* and *Thy1* in Figs. 4 and 6, three independent biological replicates were stained and scored at each timepoint. All error bars throughout the manuscript are SEM. For all box plots, center line is the median, the interquartile range is 25th–75th percentile, and outliers are eliminated. Data randomization was used to subset cells from snRNA-seq data for Supplementary Fig. 2d. No statistical method was used to predetermine size. No data were excluded from analyses. Investigators were not blinded during data collection or analysis.

## Reporting summary

Further information on research design is available in the Nature Research Reporting Summary linked to this article.

# Data availability

All sequencing data generated in this study have been deposited to GEO database under accession code GSE198767. Published data used in this study: GSE150538, GSE64825, GEO: GSE161621, GSE167597. Source Data are provided with this paper.

# Code availability

Code used in this study is available on GitHub [https://github.com/gifford-lab/motor-neuron-maturation].

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

## Acknowledgements

We thank Oliver Hobert and Emily Lowry for their feedback on the manuscript. We also thank all members of the Wichterle, Gifford, Zhang, and Au labs for their feedback on the project. We thank our funding sources: T.P. was funded by NIH T32 MH15174-39, NINDS Postdoctoral NRSA Fellowship F32NS105372, and Brain Initiative K99 (K99NS121136). J.H. was funded by NSF Graduate Research Fellowship (1122374). This work was supported by R01NS109217 (H.W. and D.K.G.), R01NS116141 (H.W.), and Project ALS (H.W.).

## Author contributions

T.P. and H.W. conceptualized and designed the project; T.P. and S.A. performed all experiments with the help from T.L.M.; all genomic data were processed and analyzed by J.H. and T.P. with the help from S.J. and M.C. and feedback from D.K.G. and H.W.; J.A.B. contributed snRNA-seq data before publication. T.P. wrote the first draft of the manuscript. T.P. and H.W. edited the manuscript with input from all authors.

## Competing interests

The authors declare no competing interests.
