## [Peer Review File · Nature Communications]

Transcriptional dynamics of murine motor neuron maturation in vivo and in vitroREVIEWER COMMENTS

Reviewer #1 (Remarks to the Author):

This study addresses a fundamental question in the field: how do post-mitotic neurons mature over time *in vivo* and *in vitro*? Uncovering the molecular mechanisms underlying neuronal maturation has significant implications not only for the developmental neuroscience field, but also for *in vitro* disease modelling and drug screening efforts. First, Patel et al. generate – *in vivo* – an impressive temporal trajectory of gene expression (RNA-Seq) and chromatin accessibility (ATAC-Seq) data for mouse spinal motor neurons from initial specification into aging. From this data, they found that motor neuron gene expression is dynamic until the third postnatal week and then stabilizes in the adult. Consistently, the ATAC-Seq analysis showed that regulatory regions (primarily distal ones) are also dynamic until P13. Motif enrichment analysis identified putative regulators potentially acting in a progressive manner to achieve a mature transcriptional state. Guided by the *in vivo* data, the authors also assessed the maturation status of *in vitro* generated motor neurons (co-cultured with astrocytes). Through RNA-Seq at DIV0, DIV7 and DIV 28, they observed that similar to their *in vivo* counterparts, *in vitro* motor neurons display dynamic gene expression. Importantly, the integration of *in vitro* with *in vivo* data suggest that the *in vivo* maturation program is composed of at least two subprograms. Lastly, the finding that neuronal activity has a minor role on motor neuron maturation (at the level of gene expression and chromatin accessibility) is a very interesting point. Overall, this is a tour-de-force effort, the study is carefully executed, and the authors provide a large amount of data that support their conclusions. I have the following comments.

1. Which part of the spinal cord was used for the *in vivo* experiments at all 7 time points? This is not mentioned in Methods, but it should be highlighted in beginning of Results because it is important to know whether the *in vivo* transcriptional and open chromatin trajectories belong to motor neurons from a particular region (e.g., brachial, thoracic) or to all motor neurons of the entire spinal cord. Along the same lines, the authors should clarify in the text the putative identity of their *in vitro* generated motor neurons. Are they of cervical, brachial, or thoracic identity?
2. Re the putative regulators of maturation, do the authors think that they act in ALL spinal motor neurons, or subsets of them? They nicely show that expression of the activity-dependent gene *Fos* is detected in more motor neurons (~ 25% at P4 to ~75% at P56), suggesting that this component of the maturation program (i.e., activity-dependent TFs) is present in most (if not all) spinal motor neurons. However, this is not clear for the other putative regulators of maturation: *Nfia*, *Nfib*, *Nr3c1*, *Nr3c2*, *Pgr* and *Ar*. This is important because the bulk nuclei RNA-Seq approach cannot resolve whether most or a small fraction of spinal motor neurons express *Nfia*, *Nfib*, *Nr3c1*, *Nr3c2*, *Pgr* and *Ar*. To address this, the authors tested expression of *Nfia* and *Nfib* in Extended Data Figure 7f, but no quantification (% of motor neurons) is provided for these stainings. They state that *Nfia* is seen only in a subset of motor neurons. First, these findings are very important and should be discussed either in Results or Discussion. Second, the paper could be strengthened if the same analysis (staining or interrogation of Blum et al 2021 data) is extended to *Nr3c1*, *Nr3c2*, *Pgr* and *Ar*.
3. The authors should consider citing Allaway et al., Nature 2021 (PMID: 34552240) and Zhang et al. Elife 2022 (PMID: 35471146) because dynamic regulatory regions (primarily distal) have been reported in the development of cortical interneurons and serotonergic neurons. Also, these studies suggested, like in this paper, that changes in chromatin accessibility are a better indicator of neuronal maturation than changes in gene expression.
4. Line 382-386: A very recent study provides support to the conclusion "...*in vivo* maturation is additionally controlled by Hox factors", and this study should be cited (Catela et al., eLife, 2022, PMID: 35315772)

5. The paper has a lot of data and is lengthy. As such, this can distract the reader from the main conclusions. The authors should consider reducing the size of the manuscript, either by being less verbose, and/or removing analyses that are tangential, but not absolutely necessary. For example, the gene expression comparison between motor neurons and inhibitory neurons (Extended Data Fig 3, panel a) could be left out possibly for a future paper.

6. The data on the contribution of neuronal activity (only modest effects observed) are very intriguing. In Discussion, the authors could also mention the limitations of the in vitro system that could lead to such modest effects (i.e., a motor circuit is not present).

7. Minor comment on Lines 342-344: "upregulation to synaptic genes in culture is similar to in vivo". Because there is no muscle in these cultures, it could benefit the reader to discuss what kind of synapses in vitro motor neurons are able to make".

Reviewer #2 (Remarks to the Author):

The study by Patel et al. mapped the transcriptional profiles and chromatin accessibility of isolated motor nuclei in vivo and in vitro throughout development. They identified potential regulators of motor neuron maturation and used their well-characterized in vitro model to manipulate maturation. By comparing their bulk RNAseq data from motor neuron nuclei to previous sequencing data from cortical neurons, they find that genes upregulated during maturation are largely cell-type specific and likely to contribute to diverse neuronal functions. Patel et al. also demonstrate that while distinct motor pools are specified early during embryonic development, different motor neuron subtypes (alphas vs gammas) are specified later in postnatal life (ED Fig.3). Using ATAC-seq to explore chromatin dynamics, Patel et al. find that upregulated genes are preferentially surrounded by newly accessible chromatin peaks, which may control their maturation dependent expression. Motif analysis of these open chromatin regions throughout development identified NFI transcription factors, steroid hormone receptors, AP-1, and Mef2 transcription factors as candidate regulators of neuron maturation.

To compare gene expression changes and chromatin dynamics between in vivo and in vitro models, Patel et al. performed bulk RNA and ATAC sequencing on mouse embryonic stem cell-derived motor neurons. They found a core set of maturation genes that change expression in the same direction both in vivo and in vitro and determined that this set of genes is largely motor neuron specific.

The idea to provide an in-depth understanding of what exact type of cell is being studied when using mouse ESC-derived neurons is compelling. Identifying what developmental stage they most closely resemble and how much they reflect motor neurons in vivo is a critical step for validation and optimization of the in vitro platform. This work therefore potentially provides a valuable resource for developing stem cell-derived neuronal models for studying adult-onset disorders. Although this study was thoroughly performed, the figures clear, and the text well written, there are a number of concerns with the study design that must be addressed.

The fundamental problem with this study is the use of the Chat-Cre mouse for the isolation of motor neurons for bulk sequencing. A previous study (Alkaslasi, Piccus et al., Nature Communications 2021) used the same mouse to perform single nucleus RNA sequencing on motor neurons, and found that this line labels much more than just motor neurons. They found a large number of non-Chat-expressing neurons, perhaps due to developmental expression of Chat driving GFP expression. Furthermore, through single nucleus RNA-seq (in the same study and a similar study from Blum et al., 2021), the Chat-Cre line is shown to label visceral motor neurons (pre-ganglionic cells) and a large

variety of cholinergic interneurons. This calls into question the specificity of the Chat-Cre line to motor neurons. So while the data are compelling and could prove to be a useful resource, the possibility and extent to which their findings are driven by other cell types must be addressed.

In lines 107-113, Patel et al. claim that the Chat-Cre driver labels “relatively pure motor neurons” with some V0c Pitx2+ interneurons, which they confirm by looking at motor neuron and interneuron transcription factors (ED Fig. 1). This does not however preclude the possibility that other cell types are included in this dataset. For instance, they could look for markers of excitatory and inhibitory cell types (as the 2021 study found labeled by this Cre line), as well as markers for preganglionic neurons and interneurons from spinalcordatlas.org which pools data from 2 single nucleus RNA sequencing studies of these neurons. From a brief search in Table 2, several genes listed as enriched in “motor neurons” were characterized by Blum et al and Alkaslasi et al to be top markers of interneurons or pre-ganglionic cells.

It is possible that Patel et al. does not see the same wide expression pattern in the Chat-Cre mouse crossed to the Sun1-GFP nuclear reporter, but then they should provide evidence of this. For instance, in Figure 1c they show expression from this mouse in the ventral horn of a P21 mouse. They should show images of whole sections of each spinal cord level (brachial, thoracic, lumbar) at P21 as well as E13.5 so as to demonstrate the specificity of their mouse line.

If whole spinal cord sections reveal that their Chat-Cre line indeed labels additional cell types (including pre-ganglionic cells and multiple interneuron populations), they need to acknowledge the limitations of their bulk sequencing approach. They will also need to estimate what proportion of captured nuclei did not belong to motor neurons, as expected per Blum et al 2021 and Alkaslasi et al 2021. Perhaps they could attempt a pseudo-bulk analysis of that combined single nucleus dataset and compare it to their own bulk-seq data.

In this study, Patel et al. map the transcriptomes of motor neurons from E10.5 to 2 years of age, finding that they are most dynamic for 4 weeks, from E10.5 to P21. To do this, they isolate GFP+ nuclei from an Hb9-GFP mouse at E10.5, and from a Chat-Cre nuclear reporter mouse for all other ages. They use the different mouse lines because Chat is not expressed until E13.5 and Hb9 is not expressed in adulthood. Though this is an understandable limitation, the use of a different mouse line likely explains their data showing that MNs at E10.5 are more distinct than at all other ages (Figs. 1e, g, h). Especially given the issues outlined above with the Chat-Cre line used for the later ages, all differences between E10.5 and the other ages could be explained by the isolation of different sets of neurons by these two lines. To address this, the authors might consider performing RNA sequencing of the Hb9-GFP nuclei at E13.5 or later (if possible) and compare transcripts to Chat-Cre nuclei at the same age to confirm that these mouse lines label comparable nuclei.

Less major: Having found that activity-dependent AP-1 motifs are highly enriched both in vitro and in vivo, the authors use their well-characterized in vitro model to assess the role of activity in motor neuron maturation. To do this, they block activity in their ESC-derived MNs in vitro with tetrodotoxin (TTX) at DIV28 and perform RNAseq and ATACseq. They then compare DIV28+TTX to DIV0, DIV7, and DIV28 (without TTX), and find that activity only has a modest effect, and that “this effect is restricted to the later stages of maturation” (line 512). It is unclear how they come to this conclusion by only examining DIV28+TTX,, and they may have seen a more than modest effect if they had iteratively performed this experiment at earlier stages. The authors don’t describe why they chose the DIV28 time point for this manipulation, and it seems like a strange choice. By their characterization of the in vitro system, DIV28 most closely resembles P13/P21 (line 363, Fig 5a, b). I would expect that activity would play a more important role in MN maturation at earlier stages, for instance around the time of developmental pruning of MNs. The rationale behind this experiment should be more clearly discussed.

Minor concerns:

Please add the gating parameters used for the FACS experiments in a supplementary figure to better understand the selection of GFP+ nuclei.

Figure 1d: How were these interneurons transcription factors selected? Markers for cholinergic interneurons in adults were thoroughly described by Blum et al. and Alkaslasi et al. and could be investigated here.

Line 166: The finding that there is a MN-specific maturation signature is compelling and exciting, but I wonder why they compared their data only to cortical neurons, when there are developmental sequencing data of other spinal cord neurons (for instance Rosenberg et al. 2018 and Delile et al. 2019). It would be interesting to see this maturation signature compared to other spinal cord cell types, otherwise the use only of cortical neuron data should be justified.

ED Fig. 3d: How were these genes for alpha and gamma MNs selected? If from Blum et al., are these genes specific to alpha or gamma as a whole, or to subtypes of alpha and gamma? If the bulk data here contains cell types other than alpha and gamma MNs, these genes would have to be specific to alpha and gamma MNs even amongst non-MNs for this analysis to be valid.

Line 190: Alkaslasi, Piccus et al. 2021 should also be cited.

Line 370: In an effort to identify regulators of genes upregulated in both their in vivo and in vitro systems, the authors performed a motif enrichment analysis on sites proximal to these genes, despite the statement on lines 213-216 that "dynamic changes in gene expression are regulated by distal regulatory regions rather than promoter proximal ones" (in fact, they say ~95% of differentially accessible chromatin regions are distal). This seems to support that they should have performed this analysis on the distal sites (~95%), rather than proximal (~5%).

Figure 3f, g: The authors show that Fos isn't detected in their data, but is detectable in the tissue. In 3f, they show that Fos is detected at E10.5, but not at the later ages. Then, in 3g, they show Fos detected at P4 and P56, but they don't show E10.5. They should show Fos expression at E10.5. Given the different mouse line used at E10.5, it's possible that Fos wasn't detected in the sequencing data at the later ages because of a mouse-specific issue.

Line 547: Include Jax ID #s for all mouse lines used.

Reviewer #3 (Remarks to the Author):

In this manuscript Patel, Wichterle and colleagues, map the transcriptional trajectory of mouse spinal motor neurons (MNs) in vivo through various stages during development and aging using RNA-Seq and ATAC-Seq. Intriguingly, they find that the expression pattern of MNs becomes fully mature by P21 and is relatively stable and maintained up to 2 years. They then compare their findings with temporal profiling of stem cell derived MNs differentiated in vitro and identified a core set of 16 genes that recapitulate the in vivo maturation program. They also find that neuronal activity mostly affects the later phase of transcriptional maturation. Collectively, their data suggest 2 distinct transcriptional phases driven intrinsically and by extrinsic factors driving specification and maturation respectively.

The study provides a comprehensive dataset of the transcriptional regulators that govern MN identity maturation and aging and as such it will serve as an invaluable

resource for the community. The experiments are well-designed, carefully controlled and executed.

The work is critically important for the field as it furthers our understanding of the transcriptional programs controlling neuronal identity and development and further supports the fidelity of in vitro differentiation programs. The manuscript should be published upon some revisions.

Major concerns:

--As it stands, the work is rather descriptive and it would substantially benefit by testing the hypothesis proposed i.e., that certain transcriptional regulators (Hox and hormone signaling factors?) can facilitate the maturation or ageing of stem cell derived MNs. This experiment could serve as proof-of-principle of an ability to manipulate the developmental state of cultured neurons and would significantly substantiate the conclusions of the paper by providing experimental evidence.

-- the TTX experiments are very interesting yet the strong conclusions i.e. "Together these data suggest that activity dependent transcription factors function as modulators, rather than drivers of the neuronal maturation program" would be better supported by an additional experiment that would assess whether enhancing neuronal activity in vitro can drive the neuronal maturation program. This can be tested through e.g., a consistent optogenetic stimulation

Minor concerns:

--The authors used whole MNs from E10.5 animals, while they used MN nuclei for the rest of the work. While this difference is not concerning regarding the ATAC-seq data, comparison of RNA-seq between whole cells and nuclei is problematic. The authors provide a panel of the expression of several genes known to be dynamic between E10.5 and E13.5 that somewhat alleviates this concern (Extended Fig. 1a). However, differences can still arise from the presence of RNA in the cytoplasm only in one of the two datasets. The authors should discuss this possibility when they describe the differential expression of 40% of the genes between E10.5 and E13.5 in Fig 1f/g. Moreover, when describing the enriched pathways of the up- or down-regulated genes in Ext. figure 1c and d, it would be preferred to compare E13.5 to P21, to avoid this confounding element.

--The age-dependent downregulated genes and pathways could be insightful regarding age-dependent neurodegenerative processes that occur in ALS --- this should be discussed

--For the in vitro stem cell differentiations the authors co-cultured MNs with astrocytes, presumably because that is the best way to mature them. A simple experiment what would add value to the study would be to directly test this hypothesis by assessing MNs without astrocytes for the key transcriptional programs they have identified. Are astrocytes one of the extrinsic factors that drive the program? Also, a comment on the dissection between astrocytic and MN transcripts in the analysis is also warranted

Response to Reviewers

We thank the reviewers for their supportive and constructive comments that helped to significantly improve the revised manuscript. We addressed most of the raised concerns in the revision and below we provide a detailed point-by-point response to the reviewers' specific comments.

REVIEWER COMMENTS

Reviewer #1 (Remarks to the Author):

This study addresses a fundamental question in the field: how do post-mitotic neurons mature over time *in vivo* and *in vitro*? Uncovering the molecular mechanisms underlying neuronal maturation has significant implications not only for the developmental neuroscience field, but also for *in vitro* disease modelling and drug screening efforts. First, Patel et al. generate – *in vivo* – an impressive temporal trajectory of gene expression (RNA-Seq) and chromatin accessibility (ATAC-Seq) data for mouse spinal motor neurons from initial specification into aging. From this data, they found that motor neuron gene expression is dynamic until the third postnatal week and then stabilizes in the adult. Consistently, the ATAC-Seq analysis showed that regulatory regions (primarily distal ones) are also dynamic until P13. Motif enrichment analysis identified putative regulators potentially acting in a progressive manner to achieve a mature transcriptional state. Guided by the *in vivo* data, the authors also assessed the maturation status of *in vitro* generated motor neurons (co-cultured with astrocytes). Through RNA-Seq at DIV0, DIV7 and DIV 28, they observed that similar to their *in vivo* counterparts, *in vitro* motor neurons display dynamic gene expression. Importantly, the integration of *in vitro* with *in vivo* data suggest that the *in vivo* maturation program is composed of at least two subprograms. Lastly, the finding that neuronal activity has a minor role on motor neuron maturation (at the level of gene expression and chromatin accessibility) is a very interesting point. Overall, this is a tour-de-force effort, the study is carefully executed, and the authors provide a large amount of data that support their conclusions. I have the following comments.

1. Which part of the spinal cord was used for the *in vivo* experiments at all 7 time points? This is not mentioned in Methods, but it should be highlighted in beginning of Results because it is important to know whether the *in vivo* transcriptional and open chromatin trajectories belong to motor neurons from a particular region (e.g., brachial, thoracic) or to all motor neurons of the entire spinal cord. Along the same lines, the authors should clarify in the text the putative identity of their *in vitro* generated motor neurons. Are they of cervical, brachial, or thoracic identity?

We apologize for not stating the regional identity of analyzed motor neurons in the results section (it was mentioned in the Methods). As our study focuses on general shared, rather than subtype-specific programs of motor neuron maturation we combined motor neurons of different columnar and pool identities in the bulk analysis. We provide this clarification in Results, section 1 in lines 120, 132 for *in vivo* spinal cords and line 402 for differentiated

motor neurons. This is an important point, as *in vitro* differentiated neurons predominantly have cervical identity, while the *in vivo* RNA-seq data includes both cervical and lumbar motor neurons. For this reason, we verified that the difference between *in vivo* and *in vitro* maturation is not driven by genes expressed in subtypes that are missing *in vitro*. If the *in vivo*-specific maturation program was overrepresented for subtype-specific genes, one would expect this set of genes to exhibit higher variance in their expression levels across distinct motor neuron subtypes identified by snRNA-seq (Blum et al., 2021). In contrast, we see that the *in vivo*-specific genes show a lower variance across different motor neuron subtypes compared to maturation genes shared between both *in vivo* and *in vitro* systems (distribution of variance shown to the right). We also examined the expression of *in vivo* specific genes across motor neuron subtypes in Blum et al., and did not find enrichment of these genes in any specific subtypes (not shown). Thus, we conclude that the majority of *in vivo*-specific maturation genes do not belong to specific subtypes that may be missing *in vitro*.

2. Re the putative regulators of maturation, do the authors think that they act in ALL spinal motor neurons, or subsets of them? They nicely show that expression of the activity-dependent gene Fos is detected in more motor neurons (~ 25% at P4 to ~75% at P56), suggesting that this component of the maturation program (i.e., activity-dependent TFs) is present in most (if not all) spinal motor neurons. However, this is not clear for the other putative regulators of maturation: Nfia, Nfib, Nr3c1, Nr3c2, Pgr and Ar. This is important because the bulk nuclei RNA-Seq approach cannot resolve whether most or a small fraction of spinal motor neurons express Nfia, Nfib, Nr3c1, Nr3c2, Pgr and Ar. To address this, the authors tested expression of Nfia and Nfib in Extended Data Figure 7f, but no quantification (% of motor neurons) is provided for these stainings. They state that Nfia is seen only in a subset of motor neurons. First, these findings are very important and should be discussed either in Results or Discussion. Second, the paper could be strengthened if the same analysis (staining or interrogation of Blum et al 2021 data) is extended to Nr3c1, Nr3c2, Pgr and Ar.

We thank the reviewer for this suggestion and have added stainings as well as scoring for Nfia, Nfib, and steroid hormone receptors for which antibodies worked in our hands (Nr3c1 and Nr3c2) to **Figure 3** and to the text (starting at line 345). We performed this analysis in E13.5, P4, and P56 ventral horn skeletal motor neurons. Nfia and Nfib are broadly expressed in large adult motor neurons but are dim or off in smaller motor neurons which are most likely to be gamma motor neurons based on Blum et al., 2021. We now discuss this in results line 349.

Immunostaining revealed that both Nr3c1 and Nr3c2 are expressed in nearly all adult skeletal motor neurons (**Figure 3g**). Interestingly, while Nr3c1 and Nr3c2 mRNA is increased already by E13.5 and P4, respectively; proteins only become visible in postnatal P4 and P56

neurons, respectively, consistent with the timing of their motif enrichment in our ATAC-seq data (line 382). Thus, the immunofluorescence analysis provides further support to the observed maturation dependent increase in accessibility of enhancers enriched for the NR motifs.

3. The authors should consider citing Allaway et al., Nature 2021 (PMID: 34552240) and Zhang et al. Elife 2022 (PMID: 35471146) because dynamic regulatory regions (primarily distal) have been reported in the development of cortical interneurons and serotonergic neurons. Also, these studies suggested, like in this paper, that changes in chromatin accessibility are a better indicator of neuronal maturation than changes in gene expression.

We have added these citations at line 294.

4. Line 382-386: A very recent study provides support to the conclusion "...in vivo maturation is additionally controlled by Hox factors", and this study should be cited (Catela et al., eLife, 2022, PMID: 35315772)

We also added this citation at line 528.

5. The paper has a lot of data and is lengthy. As such, this can distract the reader from the main conclusions. The authors should consider reducing the size of the manuscript, either by being less verbose, and/or removing analyses that are tangential, but not absolutely necessary. For example, the gene expression comparison between motor neurons and inhibitory neurons (Extended Data Fig 3, panel a) could be left out possibly for a future paper.

We believe that the comparison with cortical cell types is of broad interest to the community. It makes the important point that maturation genes upregulated in cultured motor neurons do not represent a generic, nervous-system-wide maturation program, but rather a motor neuron-specific one. For these reasons, we would like to keep this section in the revised manuscript.

At the same time, we have shortened and simplified the text at other places.

6. The data on the contribution of neuronal activity (only modest effects observed) are very intriguing. In Discussion, the authors could also mention the limitations of the *in vitro* system that could lead to such modest effects (i.e., a motor circuit is not present).

We agree that *in vivo* and *in vitro* circuits are quite different, but despite these differences, our data demonstrate that most AP-1 motif-containing enhancers are shared between the *in vivo* and *in vitro* motor neurons. This conclusion is supported by the following observations:

1. Maturation dependent accessible regions that are shared between *in vivo* and *in vitro* neurons are enriched for AP-1 motifs, suggesting that AP-1 factors are important contributors to motor neuron maturation both *in vivo* and *in vitro*.
2. *In vivo*-specific accessible regions that are present in P21 neurons but not in DIV28 neurons do not show enrichment for AP-1 motifs, nor do chromatin regions surrounding genes that are up-regulated *in vivo*, but not *in vitro*. This suggests that *in*

in vivo specific accessibility and gene expression program is not driven by qualitatively different utilization of activity regulated transcription factors.

The results section now includes the following modifications to emphasize the difference in the circuits at line 522:

“Interestingly, we have not detected enrichment for AP-1 motifs in in vivo-specific accessible regions, indicating that most activity-dependent chromatin changes are recapitulated in cultured stem cell-derived motor neurons. Together these data suggest that, despite the differences in neural circuits formed in vivo and in vitro, activity and NFI factors are important contributors to motor neuron maturation in both systems. In vivo motor neuron maturation appears to be additionally controlled by Hox factors and hormonal signaling in postnatal animals⁸².”

We agree that the modest effect of neuronal activity on motor neuron maturation is intriguing and surprising. However, we would like to draw attention to the fact that despite their conclusions that activity is a major driver of maturation in neocortex, Stroud et al. (Neuron, 2020) also state that the effects of suppression of neuronal activity in Sst interneurons has only a modest effect on gene expression: *“Between KIR2.1- and KIR2.1mut-expressing nuclei, we identified 264 genes that are significantly downregulated in KIR2.1 compared to KIR2.1mut ($p < 0.05$). We note that the degree of reduction in gene expression upon KIR2.1 overexpression is modest”* (p. 13, and also see their Supplementary Figure 7). Thus our data appear to be in full agreement with the published data, despite the difference in the interpretation.

7. Minor comment on Lines 342-344: “upregulation to synaptic genes in culture is similar to *in vivo*”. Because there is no muscle in these cultures, it could benefit the reader to discuss what kind of synapses *in vitro* motor neurons are able to make”.

Regrettably, little is known about specific types of synapses formed in these cultures. A previous study showed that *in vitro* motor neurons can form functionally relevant microcircuits when co-cultured with appropriate interneurons (Hoang et al., Neuron 2018) and that embryoid bodies become physiologically active, exhibiting rhythmic calcium activity (Sternfeld et al., eLife 2017). Since the identity and diversity of motor neuron synaptic targets in these cultures remains unknown, we do not believe it would be productive to speculate about the role of specific synapses on the motor neuron maturation program. At the same time, we find it interesting that despite the differences in cellular context, largely overlapping categories of synaptic genes and similar AP-1 containing enhancers are co-regulated in cultured and *in vivo* neurons.

Reviewer #2 (Remarks to the Author):

The study by Patel et al. mapped the transcriptional profiles and chromatin accessibility of isolated motor nuclei *in vivo* and *in vitro* throughout development. They identified potential regulators of motor neuron maturation and used their well-characterized *in vitro* model to manipulate maturation. By comparing their bulk RNAseq data from motor neuron nuclei to previous sequencing data from cortical neurons, they find that genes upregulated during maturation are largely cell-type specific and likely to contribute to diverse neuronal functions. Patel et al. also demonstrate that while distinct motor pools are specified early during

embryonic development, different motor neuron subtypes (alphas vs gammas) are specified later in postnatal life (ED Fig.3). Using ATAC-seq to explore chromatin dynamics, Patel et al. find that upregulated genes are preferentially surrounded by newly accessible chromatin peaks, which may control their maturation dependent expression. Motif analysis of these open chromatin regions throughout development identified NFI transcription factors, steroid hormone receptors, AP-1, and Mef2 transcription factors as candidate regulators of neuron maturation.

To compare gene expression changes and chromatin dynamics between in vivo and in vitro models, Patel et al. performed bulk RNA and ATAC sequencing on mouse embryonic stem cell-derived motor neurons. They found a core set of maturation genes that change expression in the same direction both in vivo and in vitro and determined that this set of genes is largely motor neuron specific.

The idea to provide an in-depth understanding of what exact type of cell is being studied when using mouse ESC-derived neurons is compelling. Identifying what developmental stage they most closely resemble and how much they reflect motor neurons in vivo is a critical step for validation and optimization of the in vitro platform. This work therefore potentially provides a valuable resource for developing stem cell-derived neuronal models for studying adult-onset disorders. Although this study was thoroughly performed, the figures clear, and the text well written, there are a number of concerns with the study design that must be addressed.

1. The fundamental problem with this study is the use of the Chat-Cre mouse for the isolation of motor neurons for bulk sequencing. A previous study used the same mouse to perform single nucleus RNA sequencing on motor neurons, and found that this line labels much more than just motor neurons. They found a large number of non-Chat-expressing neurons, perhaps due to developmental expression of Chat driving GFP expression. Furthermore, through single nucleus RNA-seq (in the same study and a similar study from Blum et al., 2021), the Chat-Cre line is shown to label visceral motor neurons (pre-ganglionic cells) and a large variety of cholinergic interneurons. This calls into question the specificity of the Chat-Cre line to motor neurons. So while the data are compelling and could prove to be a useful resource, the possibility and extent to which their findings are driven by other cell types must be addressed.

In lines 107-113, Patel et al. claim that the Chat-Cre driver labels “relatively pure motor neurons” with some V0c Pitx2+ interneurons, which they confirm by looking at motor neuron and interneuron transcription factors (ED Fig. 1). This does not however preclude the possibility that other cell types are included in this dataset. For instance, they could look for markers of excitatory and inhibitory cell types (as the 2021 study found labeled by this Cre line), as well as markers for preganglionic neurons and interneurons from spinalcordatlas.org which pools data from 2 single nucleus RNA sequencing studies of these neurons. From a brief search in Table 2, several genes listed as enriched in “motor neurons” were characterized by Blum et al and Alkaslasi et al to be top markers of interneurons or pre-ganglionic cells.

It is possible that Patel et al. does not see the same wide expression pattern in the Chat-Cre mouse crossed to the Sun1-GFP nuclear reporter, but then they should provide evidence of this. For instance, in Figure 1c they show expression from this mouse in the ventral horn of a P21 mouse. They should show images of whole sections of each spinal cord level (brachial, thoracic, lumbar) at P21 as well as E13.5 so as to demonstrate the specificity of their mouse line.

If whole spinal cord sections reveal that their Chat-Cre line indeed labels additional cell types (including pre-ganglionic cells and multiple interneuron populations), they need to acknowledge the limitations of their bulk sequencing approach. They will also need to estimate what proportion of captured nuclei did not belong to motor neurons, as expected per Blum et al 2021 and Alkaslasi et al 2021. Perhaps they could attempt a pseudo-bulk analysis of that combined single nucleus dataset and compare it to their own bulk-seq data.

We agree that this is a critically important point that should have been more thoroughly examined and addressed in the original submission. We re-examined potential contaminations reported in the above mentioned single cell studies, performed the recommended analyses, and included additional images. All evidence suggests that skeletal motor neurons are the predominant cell type in our data.

As the reviewer points out, there is a possibility of contamination from two types of populations: a) non-cholinergic cells that ectopically recombined the reporter and b) cholinergic visceral motor neurons and cholinergic interneurons.

We address both these possibilities in the revised manuscript using marker genes and integrated snRNA-seq data from Blum et al. and Alkaslasi et al. manuscripts:

- a) Alkaslasi, Piccus et al., Nature Communications 2021 report a large population of non-cholinergic neurons in their single cell study. Strikingly, they found that ~53% of cells labeled by Chat-Cre were non-cholinergic (specifically, GABAergic and Glutamatergic). To determine whether our dataset also has high representation from non-cholinergic neurons, we examined the expression of cholinergic as well as GABAergic, glutamatergic, dopaminergic, and serotonergic pathway marker genes. In contrast to high level of expression of cholinergic genes (Chat, Slc5a7), we detected only a low expression of GABAergic genes (Gad1, Gad2), and the glutamatergic gene, Slc17a6, and almost no expression of dopaminergic (Th, Ddc), and serotonergic (Tph1, Tph2) genes (Extended Data Figure 2b). However, by examining the integrated snRNAseq, we find that Gad1, Gad2, and Slc17a6 are also expressed to some extent in cholinergic neurons, including skeletal motor neurons (expression plots below and dotplot in Extended Data Figure 2c), making it unclear that their low expression in our dataset is a result of contamination from non-cholinergic cell types. The lack of expression of transcription factor markers of non-cholinergic cell types (Delile et al 2019, Bikoff et al 2016) in postnatal RNA-seq data also emphasizes this point (Figure 1d). These data together suggest that contamination from non-cholinergic neurons is likely to be minimal.

Right: Expression of cholinergic (Chat), GABAergic (Gad1, Gad2) and Glutamatergic (Slc17a6) genes in skeletal motor neurons. Integrated snRNA-seq data from spinalcordatlas.org. Example regions with expression are zoomed in.

b) The reviewer points out that some of the gene expression data could be coming from cholinergic interneurons and visceral motor neurons. We used only the brachial and lumbar regions of the spinal cord in order to increase the representation of the skeletal motor neurons in the RNA-seq data, however it is true that our data could not completely exclude cholinergic interneurons and visceral motor neurons. To address the relative representation of skeletal motor neurons better, we have added extra figure panels. First, we performed a correlation analysis between our bulk adult dataset and integrated adult snRNA-seq data of cholinergic cell types (skeletal motor, visceral, and interneurons) from Blum et al. and Alkaslasi et al. To do this, we filtered expressed genes for the top ~1000 or ~top 2000 genes that show high variance between the three cholinergic cell types, and then performed a correlation analysis on the expression of these sets of genes. This analysis showed that our bulk data and skeletal motor neurons are more highly correlated to each other than to visceral or interneurons (Extended Data Fig. 2d). We have also modified Figure 1d, to show a panel of skeletal motor neuron, visceral motor neuron, and interneuron transcription factors (including ones identified in Blum et al. and Alkaslasi et al.). Last, we have included a plot showing expression of skeletal motor neuron, visceral motor neuron, and interneuron effector genes in Extended Data Fig. 2f. In addition to high expression of motor neuron genes, we note that we see low expression of a subset of interneuron/visceral motor neuron genes. We have modified the first section of the results to reflect this (starting at line 117).

“Another limitation of using the Chat-Cre driver from E13.5-2yr is that it labels all cholinergic neurons including skeletal motor neurons, visceral motor neurons, and interneurons (Extended Data Fig. 1b, 1c, 2b, 2c). In order to increase the relative abundance of skeletal motor neurons, we isolated nuclei for RNA-seq only from cervical and lumbar regions. By performing a correlation analysis of our bulk adult data (P56) with adult snRNA-seq data sets of spinal cholinergic cell types^{31,32}, we find that bulk P56 gene expression is most highly correlated to skeletal motor neurons (Extended Data Fig. 2d). We also examined the expression levels of cell type specific transcription factors and effector genes in our dataset³⁰⁻⁴¹. Whereas skeletal motor neuron transcription factors (Isl1, Ebf1, Onecut2, Stat2, Klf6, and Foxp1) and effector genes are highly expressed, cholinergic visceral/interneuron

transcription factors and effector genes are expressed at lower levels or not expressed at all (mean expression value: skeletal 559, visceral 82, interneuron 99) (**Fig. 1d, Extended Data Fig. 1e,f**). Importantly, we do not observe consistent expression of markers of non-cholinergic cell types (**Fig. 1d**). We conclude from this analysis that skeletal motor neurons are the predominant cell type in our data, with some contribution from cholinergic interneurons and visceral motor neurons. For ATAC-seq we utilized cervical, thoracic, and lumbar regions of the spinal cord (Methods), however we ensured that all candidate regulators identified from this analysis are expressed in skeletal motor neurons.”

Images of whole spinal cords:

As the reviewer suggests, we have added images of whole spinal cords from E13.5 and adult spinal cords to Extended Data Figure 1. At E13.5, neurons labeled by SUN1-GFP seem to be limited to skeletal motor neurons in the cervical and lumbar cords. The thoracic sections show some lateral visceral motor neurons as well as some intermediate zone neurons.

At the adult stage, without staining for GFP, SUN1-GFP is primarily observed in cholinergic neurons, and is brightest in motor neurons. However, if we stain for GFP, we see labeling also in non-cholinergic neurons. We note that this seems consistent with Alkaslasi et al: cholinergic cells also seem brighter in images of spinal cords shown in Alkaslasi et al., Supplementary Figure 3. It is therefore possible that the INTACT isolation of nuclei (for RNA-seq) is biased towards skeletal motor neurons expressing the reporter at higher levels compared to other cells in the cervical and lumbar regions of adult spinal cord.

Finally, we want to make two additional points:

- a. We want to emphasize that candidate regulators of maturation identified by our analysis are generally enriched in skeletal motor neurons (shown below).

- b. We also want to add that any possible contribution of interneuron or visceral neuron expression data to the bulk RNA-seq does not have a major influence on our *in vivo*-

in vitro comparisons. Genes that are upregulated both *in vivo* and *in vitro* show similar correlation between bulk P56 and snRNA-seq from skeletal motor neurons as genes that are only upregulated *in vivo* (coefficient of correlation: 0.66 and 0.67 respectively).

2. In this study, Patel et al. map the transcriptomes of motor neurons from E10.5 to 2 years of age, finding that they are most dynamic for 4 weeks, from E10.5 to P21. To do this, they isolate GFP+ nuclei from an Hb9-GFP mouse at E10.5, and from a Chat-Cre nuclear reporter mouse for all other ages. They use the different mouse lines because Chat is not expressed until E13.5 and Hb9 is not expressed in adulthood. Though this is an understandable limitation, the use of a different mouse line likely explains their data showing that MNs at E10.5 are more distinct than at all other ages (Figs. 1e, g, h). Especially given the issues outlined above with the Chat-Cre line used for the later ages, all differences between E10.5 and the other ages could be explained by the isolation of different sets of neurons by these two lines. To address this, the authors might consider performing RNA sequencing of the Hb9-GFP nuclei at E13.5 or later (if possible) and compare transcripts to Chat-Cre nuclei at the same age to confirm that these mouse lines label comparable nuclei.

As the reviewer suggests, we had previously attempted to purify motor neurons from E13.5 Hb9-gfp animals. However, in our hands, the survival of motor neuron cells at E13.5 was about 1/5th that at E10.5 after dissociation and FACs sorting, raising the concern that only more resistant motor neuron subtypes will be represented in the sorted population. Our ATACseq data from these FAC sorted cells were also of low quality in two replicates, and we therefore did not perform additional experiments with these cells.

We understand the reviewer's concerns about differences between E10.5 and E13.5 being driven by isolation of different cell types. However, we are reassured by the fact that SUN1-GFP labeling in cervical and lumbar regions is restricted to skeletal motor neurons in E13.5 spinal cords (Extended Data Fig 1a). Genes that are known to change expression in spinal motor neurons between these ages recapitulate the correct dynamic (Extended Data Fig 2a). In addition, we show here that genes that are known embryonic markers for interneurons and visceral motor neurons do not consistently increase between E10.5 and E13.5. VMN genes: *Nos1*, *Zeb2*; IN genes: *Pax2*, *Pitx2*. All of this supports the idea that expression changes between E10.5 and E13.5 are not solely driven by changes in cell types.

Another reason the comparison between E10.5 and E13.5 might be confounded is the fact that we collect whole cells at E10.5 and only nuclei from E13.5 onwards, we have addressed this concern in response to Reviewer 3, comment 3.

Expression values of embryonic markers of visceral neurons and interneurons

3. Less major: Having found that activity-dependent AP-1 motifs are highly enriched both in vitro and in vivo, the authors use their well-characterized in vitro model to assess the role of activity in motor neuron maturation. To do this, they block activity in their ESC-derived MNs in vitro with tetrodotoxin (TTX) at DIV28 and perform RNAseq and ATACseq. They then compare DIV28+TTX to DIV0, DIV7, and DIV28 (without TTX), and find that activity only has a modest effect, and that “this effect is restricted to the later stages of maturation” (line 512). It is unclear how they come to this conclusion by only examining DIV28+TTX,, and they may have seen a more than modest effect if they had iteratively performed this experiment at earlier stages. The authors don’t describe why they chose the DIV28 time point for this manipulation, and it seems like a strange choice. By their characterization of the in vitro system, DIV28 most closely resembles P13/P21 (line 363, Fig 5a, b). I would expect that activity would play a more important role in MN maturation at earlier stages, for instance around the time of developmental pruning of MNs. The rationale behind this experiment should be more clearly discussed.

In the TTX experiment, our goal was to understand if activity was required for cultured motor neurons to undergo changes in gene expression and chromatin accessibility that accompany maturation. To test this, motor neurons were cultured with TTX from DIV0 - DIV28 and then we asked if DIV28 motor neurons looked less mature. Thus activity was blocked at all stages of *in vitro* maturation, including the stage when potential motor neuron pruning might occur. We found that cells were indeed less mature in the absence of activity, and that the genes and accessible sites affected were largely those that change between DIV7-DIV28. To make this point more convincing, we have now added data from DIV7 motor neurons with and without activity (TTX treatment from DIV0 - DIV7). Consistent with our previous conclusion, we found that the presence of TTX had almost no effect on motor neuron gene expression and accessibility between DIV0 - DIV7 (Figure 6 and Extended Data Figure 10).

Minor concerns:

4. Please add the gating parameters used for the FACS experiments in a supplementary figure to better understand the selection of GFP+ nuclei.

We have added this to Extended Data Figure 1.

5. Figure 1d: How were these interneurons transcription factors selected? Markers for cholinergic interneurons in adults were thoroughly described by Blum et al. and Alkaslasi et al. and could be investigated here.

The interneuron transcription factors in this figure were previously selected from snRNA-seq of whole spinal cords performed in Delile et al 2019, and from Bikoff et al 2016 (bulk sequencing of interneuron populations at various ages). We have now added transcription factors and effector genes identified in Blum et al. and Alkaslasi et al. to Figure 1d and Extended Data Figure 2f.

6. Line 166: The finding that there is a MN-specific maturation signature is compelling and exciting, but I wonder why they compared their data only to cortical neurons, when there are developmental sequencing data of other spinal cord neurons (for instance Rosenberg et al. 2018 and Delile et al. 2019). It would be interesting to see this maturation signature compared to other spinal cord cell types, otherwise the use only of cortical neuron data should be justified.

Longitudinal studies done by us and others suggest that neuronal maturation is complete around ~P21 for a variety of cell types. We compared our dataset to cortical neurons for three reasons. The first is technical: to understand similarities and differences in maturation trajectories we used a highly comparable longitudinal dataset - Stroud et al., perform RNAseq on INTACT-purified nuclei at P7, P21, and P56 using methods very similar to ours. The second reason is that Stroud et al had an immature time point (P7) as well as fully mature time points (P21, and P56), allowing us to compare postnatal maturation trajectories. And lastly, through this analysis we wanted to see if there was a shared, nervous-system-wide maturation program. In order to answer this question, it was better to use a distant neuron type. This reasoning is discussed in the Discussion line 795 of the manuscript.

We understand the reviewer's suggestion that other spinal cord neuron types may have more similar trajectory to motor neurons, but we did not perform this analysis because Delile et al and Rosenberg et al. do not include analysis of mature spinal cord neurons (P21 or later) .

7. ED Fig. 3d: How were these genes for alpha and gamma MNs selected? If from Blum et al., are these genes specific to alpha or gamma as a whole, or to subtypes of alpha and gamma? If the bulk data here contains cell types other than alpha and gamma MNs, these genes would have to be specific to alpha and gamma MNs even amongst non-MNs for this analysis to be valid.

Alpha and gamma markers were previously selected from Blum et al. We have now modified this to include alpha and gamma markers from Alkaslasi et al. as well. These genes are specific to alpha or gamma neurons as a whole. We previously combined alpha and gamma specific genes and showed that these genes continue to be upregulated in postnatal life between P4 and P21. With the longer lists of alpha and gamma markers, our data now shows that this trend is specific to alpha motor neuron markers and is even stronger than we previously reported (Extended Data Figure 5b-e). We have changed the text to reflect this in lines 255-262. We previously reported that 37% of alpha+gamma marker genes are

upregulated during postnatal maturation. This number changes to 48% when we consider only alpha markers integrated from Blum et al., and Alkaslasi et al.

Only 5 of the alpha markers are truly specific to skeletal motor neurons (rest are expressed to some extent in visceral and interneurons as per spinalcordatlas.org). All 5 of these markers - *Aox1*, *Vipr2*, *Glis3*, *Mrv1*, *Col4a5* - are upregulated between P4-P21.

8. Line 190: Alkaslasi, Piccus et al. 2021 should also be cited.

We sincerely apologize for the oversight, and have cited Alkaslasi, Piccus et al. 2021 here and in all relevant places where analysis was modified to include their data.

9. Line 370: In an effort to identify regulators of genes upregulated in both their in vivo and in vitro systems, the authors performed a motif enrichment analysis on sites proximal to these genes, despite the statement on lines 213-216 that “dynamic changes in gene expression are regulated by distal regulatory regions rather than promoter proximal ones” (in fact, they say ~95% of differentially accessible chromatin regions are distal). This seems to support that they should have performed this analysis on the distal sites (~95%), rather than proximal (~5%).

The analysis was indeed performed on the distal sites which make up 95% of the differentially accessible regions. In this context, by proximal we meant distal accessible regions (i.e. >2kb away from TSS) that are most proximal to the genes being considered. We have changed the language in lines 509-511 to clarify this point:

“We performed motif enrichment analysis on distal genomic sites that are associated with maturation dependent genes and that gain accessibility over time both in vivo and in vitro (between E10.5-P21 in vivo and between DIV0-DIV28 in vitro).”

10. Figure 3f, g: The authors show that Fos isn't detected in their data, but is detectable in the tissue. In 3f, they show that Fos is detected at E10.5, but not at the later ages. Then, in 3g, they show Fos detected at P4 and P56, but they don't show E10.5. They should show Fos expression at E10.5. Given the different mouse line used at E10.5, it's possible that Fos wasn't detected in the sequencing data at the later ages because of a mouse-specific issue.

It is true that a low level of Fos RNA is detected at E10.5 in our dataset. However, we do not observe Fos protein by immunostaining at E10.5 (not shown). As shown in the modified Figure 3, Fos begins to be detected in a small percent of motor neurons (<10%) at E13.5 and gradually increases over time. We do acknowledge that the inability to detect Fos may be because of the limitations of nuclear RNA-seq (line 371):

“Fos transcripts may not be effectively detected in bulk nuclear RNA, as this immediate early gene is only transiently and rapidly induced in response to activity”

11. Line 547: Include Jax ID #s for all mouse lines used.

We have now included these in Methods line 888.

Reviewer #3 (Remarks to the Author):

In this manuscript Patel, Wichterle and colleagues, map the transcriptional trajectory of mouse spinal motor neurons (MNs) *in vivo* through various stages during development and aging using RNA-Seq and ATAC-Seq. Intriguingly, they find that the expression pattern of MNs becomes fully mature by P21 and is relatively stable and maintained up to 2 years. They then compare their findings with temporal profiling of stem cell derived MNs differentiated *in vitro* and identified a core set of 16 genes that recapitulate the *in vivo* maturation program. They also find that neuronal activity mostly affects the later phase of transcriptional maturation. Collectively, their data suggest 2 distinct transcriptional phases driven intrinsically and by extrinsic factors driving specification and maturation respectively.

The study provides a comprehensive dataset of the transcriptional regulators that govern MN identity maturation and aging and as such it will serve as an invaluable resource for the community. The experiments are well-designed, carefully controlled and executed. The work is critically important for the field as it furthers our understanding of the transcriptional programs controlling neuronal identity and development and further supports the fidelity of *in vitro* differentiation programs. The manuscript should be published upon some revisions.

Major concerns:

1.--As it stands, the work is rather descriptive and it would substantially benefit by testing the hypothesis proposed i.e., that certain transcriptional regulators (Hox and hormone signaling factors?) can facilitate the maturation or ageing of stem cell derived MNs. This experiment could serve as proof-of-principle of an ability to manipulate the developmental state of cultured neurons and would significantly substantiate the conclusions of the paper by providing experimental evidence.

We agree that testing additional transcriptional regulators is an important next step, however we believe that it is beyond the scope of this manuscript. In this manuscript we focused on establishing the maturation trajectory *in vivo*, determining how far it can be recapitulated *in vitro*, and establishing the utility of the *in vitro* system by experimentally testing the dominant hypothesis in the field that activity is a major regulator of postnatal neuronal maturation. These experiments have led to novel findings that we think will be of general interest to the community.

Systematic and comprehensive testing of motifs discovered in maturation dependent accessible regions is not a trivial undertaking as transcription factor families predicted to bind these motifs are complex, with numerous splice variants, and diverse maturation-dependent posttranslational modifications that can influence their functionality.

2-- the TTX experiments are very interesting yet the strong conclusions i.e. "Together these data suggest that activity dependent transcription factors function as modulators, rather than drivers of the neuronal maturation program" would be better supported by an additional experiment that would assess whether enhancing neuronal activity *in vitro* can drive the

neuronal maturation program. This can be tested through e.g., a consistent optogenetic stimulation

As the reviewer suggests, we attempted to enhance neuronal activity by performing a longitudinal repetitive optogenetic stimulation of channel rhodopsin expressing motor neurons or by pulse treatments with KCl. Regrettably, both treatments led to a high degree of stress and cell death in culture. In the cells that remained alive after pulse KCl treatments, we did not find an increase in Spp1 expression (maturation dependent gene which is regulated by activity as shown in Figure 6).

We also performed RNA-seq on DIV7 and DIV28 motor neurons that were acutely treated with KCl for 1hr or 6hr. In those cases, we saw a dramatic increase in immediate early genes such as Fos and Npas4, but we did not detect a global activation of a more mature gene expression signature in the motor neurons. We opted not to include this data in the manuscript because acute treatment with KCl cannot rule out the possibility that protracted and patterned increase in motor neuron activity would accelerate the maturation process.

Minor concerns:

3--The authors used whole MNs from E10.5 animals, while they used MN nuclei for the rest of the work. While this difference is not concerning regarding the ATAC-seq data, comparison of RNA-seq between whole cells and nuclei is problematic. The authors provide a panel of the expression of several genes known to be dynamic between E10.5 and E13.5 that somewhat alleviates this concern (Extended Fig. 1a). However, differences can still arise from the presence of RNA in the cytoplasm only in one of the two datasets. The authors should discuss this possibility when they describe the differential expression of 40% of the genes between E10.5 and E13.5 in Fig 1f/g. Moreover, when describing the enriched pathways of the up- or down-regulated genes in Ext. figure 1c and d, it would be preferred to compare E13.5 to P21, to avoid this confounding element.

We agree with the reviewer that there might be discrepancies related to whole cell vs. nuclear RNA-seq data. We have added an acknowledgement in the text line 112 that this may influence differential gene expression analyses.

*“We note that using whole cells at E10.5 vs. nuclei at E13.5 onwards may result in technical differences in gene expression values and influence differential gene expression analyses. To determine if these data are comparable, we examined the expression of genes that are known to be up or downregulated between E10.5 and E13.5 and found that these genes maintain the correct dynamics in our RNA-seq dataset (**Extended Data Fig. 2a**).”*

In Extended Data Figure 3a, we have also added pathway enrichment analysis on genes up/down-regulated between E13.5 and P21. These pathways are exactly the same with one exception: “Nuclear Receptor transcription pathway” is enriched above the threshold FDR in genes upregulated between E10.5 and P21 but not in genes upregulated between E13.5 and P21.

In addition, we show here PCA analysis of the *in vivo* data from E13.5-2yr. Even after excluding E10.5 from this analysis, the major conclusion that gene expression and chromatin accessibility is dynamic till P21 remains true.

We discussed excluding E10.5 data from the manuscript because of the technical differences, however we have decided to leave it in because we think the transition from an RXR, Sox, E-box, LIM signature at E10.5 to an NFI signature at E13.5 in the ATAC-seq data represents an important first step in the maturation of motor neurons.

4--The age-dependent downregulated genes and pathways could be insightful regarding age-dependent neurodegenerative processes that occur in ALS --- this should be discussed

We have added text in line 211 and an Extended Figure 3b, to note that Proteasome complex genes are downregulated in postnatal motor neurons, which may be relevant to misfolded protein aggregation in ALS (An et al., 2019 eLife). This was the most obviously relevant set of genes that showed dramatic decrease during maturation.

“We also noted that proteasome and TriC/CCT chaperonin complex genes are significantly downregulated during the transition from embryonic to postnatal state, which may contribute to the inability to clear misfolded proteins in ALS⁵⁵ (Extended Data Fig. 3b).”

5--For the *in vitro* stem cell differentiations the authors co-cultured MNs with astrocytes, presumably because that is the best way to mature them. A simple experiment what would add value to the study would be to directly test this hypothesis by assessing MNs without astrocytes for the key transcriptional programs they have identified. Are astrocytes one of the extrinsic factors that drive the program? Also, a comment on the dissection between astrocytic and MN transcripts in the analysis is also warranted

We did attempt this experiment, but we have not identified conditions that would allow robust survival of healthy motor neurons over the 28 day period. Human stem cell derived neurons survive better without glial support and it would be interesting to perform this experiment in that context in the future.

RNA-seq was performed on purified motor neuron nuclei (Chat-Cre; Sun1-GFP + nuclei, which are all neuronal and >90% *Isl1/2+*). We have examined potential contamination of our samples by examining expression of genes enriched in astrocytes (*Gfap*, *S100b*, *Aqp4*, and

Aldh1l1). None of these genes were detected at a meaningful level in the RNA-seq data (new Extended Data Figure 8a).

REVIEWERS' COMMENTS

Reviewer #1 (Remarks to the Author):

The authors have done an excellent job addressing all my concerns. The revised manuscript is significantly strengthened by the addition of new data. The changes in text clarified important points.

Reviewer #2 (Remarks to the Author):

I am satisfied with the revisions and response to reviewers provided by the authors, in particular the efforts to understand what subset of the bulk-RNAseq data represent skeletal MNs. The authors have done a great job addressing all my concerns.

Reviewer #3 (Remarks to the Author):

While I am disappointed that the authors did not choose to test the ability of transcriptional regulators to manipulate the developmental state of cultured neurons I accept their argument that this is not a trivial experiment and is beyond the scope of the current study. Other than this the authors have addressed most of my other comments and concerns and I am therefore satisfied.

RESPONSE TO REVIEWERS' COMMENTS

Reviewer #1 (Remarks to the Author):

The authors have done an excellent job addressing all my concerns. The revised manuscript is significantly strengthened by the addition of new data. The changes in text clarified important points.

We are pleased that the reviewer is happy with the revised version of the manuscript.

Reviewer #2 (Remarks to the Author):

I am satisfied with the revisions and response to reviewers provided by the authors, in particular the efforts to understand what subset of the bulk-RNAseq data represent skeletal MNs. The authors have done a great job addressing all my concerns.

We are pleased that the reviewer is satisfied. We have now added a non-negative least squares deconvolution analysis to quantitatively understand the cellular composition of our RNA-seq data compared to snRNA-seq data from Alaskasi et al., 2021. Results are added as Supplementary Figure 1d and they are in agreement with previous analysis showing that there is no contamination from non-cholinergic cells in our bulk RNA-seq data, and that a vast majority of the contribution comes from skeletal motor neurons (>75%), with the remaining coming mostly from Visceral motor neurons.

Reviewer #3 (Remarks to the Author):

While I am disappointed that the authors did not choose to test the ability of transcriptional regulators to manipulate the developmental state of cultured neurons I accept their argument that this is not a trivial experiment and is beyond the scope of the current study. Other than this the authors have addressed most of my other comments and concerns and I am therefore satisfied.

We are pleased that the reviewer is satisfied.